# Topology-preserving smoothing of retinotopic maps

**Yanshuai Tu** [1], **Duyan Ta** [1], **Zhong-Lin Lu** [2,3,4]*, **Yalin Wang** [1]*

**1** School of Computing, Informatics, and Decision Systems Engineering, Arizona State University, Tempe, Arizona, United States of America, **2** Division of Arts and Sciences, New York University Shanghai, Shanghai, China, **3** Center for Neural Science and Department of Psychology, New York University, New York, United States of America, **4** NYU-ECNU Institute of Brain and Cognitive Science, NYU Shanghai, Shanghai, China

* zhonglin@nyu.edu (Z-LL); ylwang@asu.edu (YW)

**Data Availability Statement:** The executable program for reproducing figures, performing analyses, and a step-by-step introduction are available on the OSF website https://osf.io/dbgkf/.

**Funding:** DT, YT, ZL, and YW were supported by the Directorate for Mathematical and Physical

## Abstract

Retinotopic mapping, i.e., the mapping between visual inputs on the retina and neuronal activations in cortical visual areas, is one of the central topics in visual neuroscience. For human observers, the mapping is obtained by analyzing functional magnetic resonance imaging (fMRI) signals of cortical responses to slowly moving visual stimuli on the retina. Although it is well known from neurophysiology that the mapping is topological (i.e., the topology of neighborhood connectivity is preserved) within each visual area, retinotopic maps derived from the state-of-the-art methods are often not topological because of the low signal-to-noise ratio and spatial resolution of fMRI. The violation of topological condition is most severe in cortical regions corresponding to the neighborhood of the fovea (e.g., < 1 degree eccentricity in the Human Connectome Project (HCP) dataset), significantly impeding accurate analysis of retinotopic maps. This study aims to directly model the topological condition and generate topology-preserving and smooth retinotopic maps. Specifically, we adopted the Beltrami coefficient, a metric of quasiconformal mapping, to define the topological condition, developed a mathematical model to quantify topological smoothing as a constrained optimization problem, and elaborated an efficient numerical method to solve the problem. The method was then applied to V1, V2, and V3 simultaneously in the HCP dataset. Experiments with both simulated and real retinotopy data demonstrated that the proposed method could generate topological and smooth retinotopic maps.

## Author summary

Retinotopic maps of human observers derived from state-of-the-art methods are often not topological because of the low signal-to-noise ratio and spatial resolution of fMRI. The proposed topological smoothing method can generate topology-preserving and smooth retinotopic maps in V1, V2, and V3 simultaneously from retinotopy fMRI data.

This is a *PLOS Computational Biology* Methods paper.

Sciences, Grant no: 1413417 to DT, YT, YW, and Grant no: 1412722 to ZL. DT, YT, and YW were supported by the National Institute on Aging, Grant no: RF1AG051710 and R21AG065942. YW was also funded by the National Institute of Biomedical Imaging and Bioengineering, Grant no: R01EB025032. YT and DT were supported by Arizona Alzheimer's Consortium. The funders had no role in study design, data collection and analysis, decision to publish, or preparation of the manuscript.

**Competing interests:** I have read the journal's policy and the authors of this manuscript have the following competing interests: YT, ZL and YW have a joint patent application, Tu, Y., Y. Wang, and Z.-L. Lu, Methods and Systems for Precise Quantification of Human Sensory Cortical Areas, U. S. Patent Application No. 63/004. 2020.

## Introduction

Retinotopic mapping, i.e., the mapping between visual input on the retina to neuronal activations in visual cortical areas, is one of the central topics in visual neuroscience [1]. Although initially discovered in neurophysiology [2], Blood Oxygenation Level Dependent (BOLD) functional magnetic resonance imaging (fMRI) provides an excellent noninvasive tool to perform *in vivo* retinotopic mapping on human observers [3,4]. The procedure typically consists of (1) recording BOLD fMRI cortical activations generated by carefully designed slowly moving visual stimuli on the retina to uniquely encode the visual space [5,6], (2) co-registering the fMRI activations with structural MRI, (3) flattening the 3D structural MRI data to obtain visual cortical surface, and (4) decoding the retinal visual coordinates underlying the observed fMRI time series at each location on the cortical surface [3,7]. In the last two decades, the development of a variety of retinotopic mapping techniques has greatly extended our understanding of visual cortical organization in normal and abnormal populations [3,8–13].

Although the exact shape of visual inputs on the retina is not preserved on the cortical surface, neurophysiology studies [14–17] have shown that the mapping preserves local neighborhood geometric relationships. More specifically, neighboring points on the cortical surface should have neighboring retinal visual coordinates [18]. We adopted the term *topology preserving (or topological)* to refer to the preservation of neighborhood relationships of the observations in the output space [19]. In this work, the topological condition or topology-preserving are not related to the surface genus, handles or holes. The concept of topological condition is illustrated in Fig 1 for retinotopic mapping. Fig 1A and 1B show an example of topological mapping that preserves the structure of the source polygon (on the cortical surface) in the target (retina) domain (Fig 1B). Fig 1A and 1C show an example of non-topological mapping that breaks the source polygon structure after mapping: $f_i$ is out of the polygon edge $\bar{f_j}\bar{f_k}$, resulting in what is called a *flipped triangle* $\Delta f_i f_j f_k$ (the orientations of the triangle are different in the source and target domains).

The topological condition is a natural consequence of the hierarchical organization of the visual system: there cannot be duplicated neuronal representations of the same retinal location within each visual area (otherwise the visual area should be further divided into more subareas). Although neurophysiological studies on animals have found that the retinotopic maps in lower visual areas are topological [2], the "raw" retinotopic maps derived from BOLD fMRI recordings are usually not because of the low signal-to-noise ratio and spatial resolution of the technology [3]. Although the quality of fMRI data has improved during the last three decades, the signal-to-noise ratio is still relatively low even with ultra-high field MRI systems [20]. A

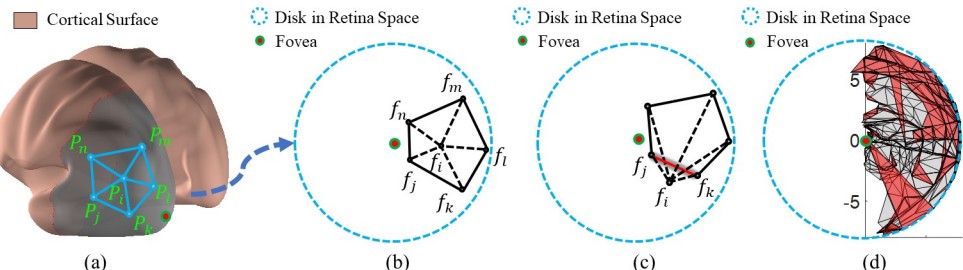

**Fig 1. Illustration of topological and non-topological maps.** (A) A polygon patch on an inflated cortical surface (source domain). (B) A topological mapping of (A) in the target domain (retina space). It preserves the structure of the source polygon. (C) A non-topological mapping of (A). It breaks the structure of the source polygon: $f_i$ is moved out of the edge $\bar{f_j}\bar{f_k}$, resulting in what is called a *flipped triangle* $\Delta f_i f_j f_k$. (d) A typical retinotopic map of V1. Flipped triangles are labeled with red color.

typical raw retinotopic map of V1 from the state of the art dataset and method [20] still contains about 20% flipped triangles (Fig 1D; Table A in S5 Text). Violations of the topological condition not only make the results from fMRI inconsistent with those from neurophysiology but also make it difficult to perform quantitative analysis on retinotopic maps.

Typically, fMRI signals are analyzed on a voxel-by-voxel basis. The most prominent method is based on the population receptive field (pRF) model, which decodes perception parameters of the fMRI signals at each voxel. However, influenced by low SNR and low spatial resolution, the retinotopic maps from the analysis are typically not topological. In the past decade, significant efforts have been devoted to improving their accuracy and stability. For instance, the reliability of perception size was discussed [12,13], and methods that could decode perception parameters by model-free back projections with potentially more precise perception shape were proposed [21–23]. Although they have greatly improved the pRF model, none of the methods has explicitly investigated the topological condition.

In addition, a number of methods have been proposed to correct or reduce topological violations in fMRI-based retinotopic maps from the pRF analysis. Yet none of them has fully solved the problem. For instance, traditional spatial smoothing methods, including Gaussian smoothing [14,24], Laplacian smoothing [25], mesh-spectrum-based smoothing [26], cannot guarantee the topological condition because retinotopic maps are defined on two-dimensional surfaces, yet these methods only smooth the maps along one dimension (e.g., polar angle) at a time. Another promising approach registers the noisy "raw" retinotopic maps to an ideal template and assigns coordinates based on the registration function/morphing [8]. Although it can generate smooth and topological retinotopic maps, the method relies on diffeomorphic registration of a noisy parametrized surface to a predefined template, which is not easy to optimize, especially when the data are noisy. Furthermore, even if the registration issue can be solved, defining the right template remains a challenging problem. Other methods based on fitting predefined algebraic models (such as Schwartz's complex "log" model [27] or Schira's "Double-Sech" model [9]) can guarantee topological condition. However, the models introduce a lot of assumptions on the data [8,28]. The topological condition was considered in our previous work [29], but it was only applied to V1 with limited validation. And, because the manually drawn boundaries were not accurate, the results were not very precise near the boundary.

We propose a topological smoothing method to generate smooth and topological retinotopic maps for multiple visual regions without assuming algebraic models. A geometric concept, Beltrami coefficient (BC) [30], is adopted to formulate the optimization problem to define, quantify, and ensure the topological condition [31]. The Beltrami coefficient is a complex number that can quantify triangle-wise geometric conformality (i.e., the angle preserving property) and topological condition for surface-to-surface mapping. When we consider the topological condition in the mapping from the polygon in Fig 1A to 1B, the overall topological condition can be checked for each triangle attached to point $P_i$. Namely, one may check whether its mapping result, $f_i$, is moved out of any of the polygon edges, $\overline{f_j f_k}$, $\overline{f_k f_l}$, $\overline{f_l f_m}$, $\overline{f_m f_n}$, and $\overline{f_n f_j}$. If $f_i$ is moved out of any of the edges, e.g., $\overline{f_j f_k}$, the mapping is not topological. Without losing generality, when considering the topological condition for polygon edge $\overline{f_j f_k}$, we can construct a coordinate system (Fig 2A and 2B). In Fig 2B, we illustrate three mapping results for $P_i$: Case (1) if $f_i$ coincides with the green dot such that triangles $\Delta P_i P_j P_k$ and $\Delta f_i f_j f_k$ are similar ($\Delta P_i P_j P_k \sim \Delta f_i f_j f_k$), the magnitude of the Beltrami coefficient associated with triangle $\Delta P_i P_j P_k$ is zero ($|\mu| = 0$). This is the case of conformal mapping. Case (2) if $f_i$ is above the $f^{(1)}$ axis ($f_i^{(2)} > 0$), shown as the yellow dot, the magnitude of the Beltrami coefficient associated with triangle $\Delta P_i P_j P_k$ is greater than zero but less than one ($0 < |\mu| < 1$). This is the case of

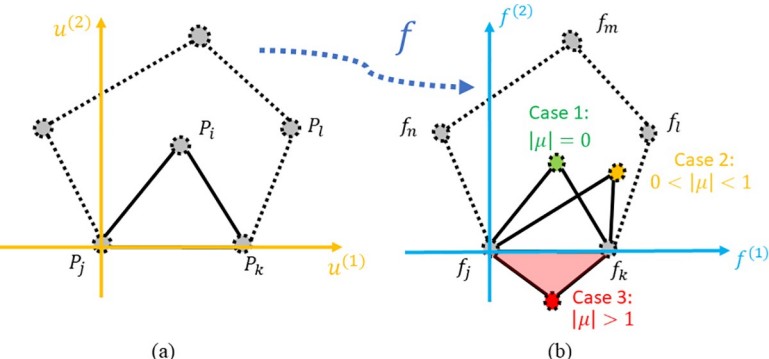

**Fig 2.** Beltrami coefficient and topological condition: (A) source triangle (triangle $\Delta P_i P_j P_k$), (B) three target triangles $\Delta f_i f_j f_k$ from three different types of mappings, with different Beltrami coefficients.

quasiconformal mapping with respect to edge $\overline{f_j f_k}$. And Case (3) if $f_i$ is below the $f^{(1)}$ axis, shown as the red dot, the magnitude of the Beltrami coefficient associated with triangle $\Delta P_i P_j P_k$ is greater than one ($|\mu|>1$) and the mapping is no longer topological. In summary, the complex-valued Beltrami coefficient encodes relative mapping information for each triangle and can be used to quantify the topological condition [32,33].

To ensure the topological condition, we defined an objective function such that, after optimally adjusting the retinal visual coordinates of each point on the cortical surface, the maximum magnitude of the Beltrami coefficients on the new retinotopic map is less than one (i.e., topological). An efficient numerical method was then developed to solve the optimization problem.

Our proposed method can work on multiple visual areas simultaneously without assuming any prior model and allow us to draw more precise boundaries between visual areas. To validate the method, we conducted experiments on both synthetic and real retinotopic map data from the Human Connectome Project (HCP) [34] and compared the performance of our method with that of existing methods based on the quality of fits to the raw fMRI time series.

## Methods

### HCP retinotopy data

The Human connectome project (HCP) [34] provides a large publicly available retinotopy dataset collected on 7T MRI scanners. The data collection, conducted on 181 healthy young adults (22–35 years; 109 females, and 72 males) with normal or corrected-to-normal visual acuity, involved carefully designed retinotopy stimuli and resulted in a substantial amount of fMRI data (30 min, 1,800 time-points) acquired at very high spatial and temporal resolutions (1.6 mm isotropic voxels, 1-second temporal sampling). The dataset provides an exciting opportunity to evaluate our method.

### Overview of the pipeline

The overall proposed processing pipeline is shown in Fig 3. In the HCP dataset, the visual stimuli consist of rotating wedges, expanding/shrinking rings, and moving bars [3,35]. The carriers of the stimuli are made of dynamic color textures that can better activate visual neurons (Fig 3A). We denote a point in the visual field by $v = (v^{(1)}, v^{(2)}) \in \mathbb{R}^2$, where $v^{(1)}$ is the eccentricity (distance to the fovea in degrees of visual angle), and $v^{(2)}$ is the polar angle relative to the positive horizontal line (Fig 3B). Both high-quality structural MRI and fMRI scans were

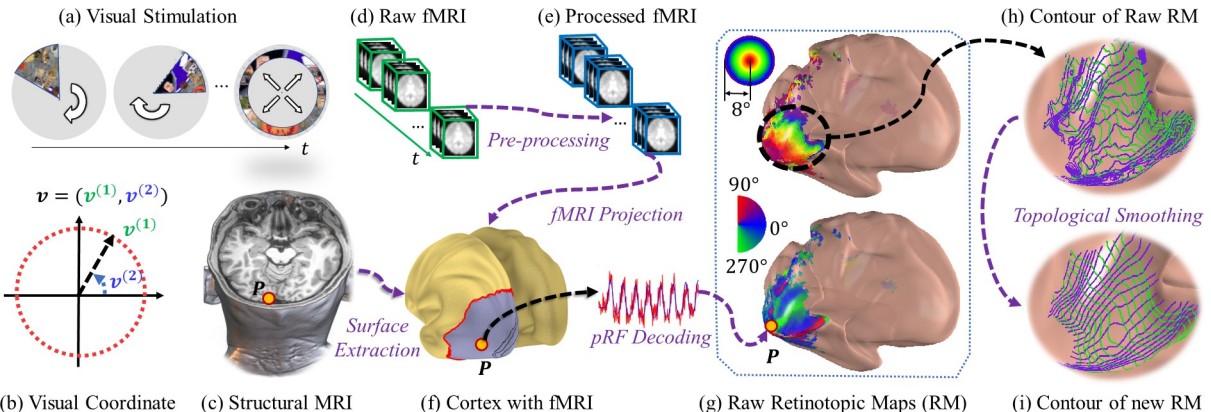

**Fig 3. The processing pipelines.** (A) Visual stimuli. (B) Coordinate system of the visual field. (C) Structural (anatomical) MR image. (D) BOLD fMRI scans. (E) Preprocessed fMRI signals. (F) Reconstructed cortical surface with projected fMRI activations. (G) Retinotopic maps from the pRF model. (H) Level set of the raw retinotopic maps. (I) Level set of the smoothed retinotopy maps.

acquired by the HCP group (Fig 3C and 3D). The processing pipeline consists of fMRI preprocessing [36], compressive population receptive field decoding [3], and topological smoothing. The raw retinotopic maps from pRF decoding (RM; Fig 3G) contain many topological violations (Fig 3H). Topological smoothing is the proposed method in this paper to fix topological violations and smooth the retinotopic maps (Fig 3I). In the following subsections, we briefly describe fMRI preprocessing and pRF decoding (although we used the publicly available decoded (raw) retinotopic maps from [20]), and then introduce the topological smoothing method. We list the key definitions and symbols in *Table A in S1 Text*.

## fMRI preprocessing

The goal of fMRI preprocessing is to detect, in a robust, sensitive, and valid way, the time series of brain activations of the voxels on the visual cortical surface that are associated with the visual stimuli. The HCP preprocessing pipeline [36] is standardized. First, a cortical surface is extracted from structural MRI using Freesurfer [37] and resampled (Fig 3F). Then, the raw fMRI data from each imaging session are co-registered across time to reduce the influence of head movements and other motion artifacts (Fig 3E). Finally, the co-registered fMRI data are projected onto the cortical surface. During the projection, spatial smoothing is applied along the cortical surface to improve the quality of the fMRI signal. The preprocessing eventually generates a resampled cortical surface as well as the fMRI activation time series on each point of the cortical surface.

## pRF decoder

We briefly describe the population receptive field model (pRF) [3,38] to introduce some necessary notations. For readers familiar with pRF model, the visual coordinates we used here are related to the x-y coordinates by $x = v^{(1)} \cos v^{(2)}, y = v^{(1)} \sin v^{(2)}$. For each voxel $P = (X,Y,Z) \in \mathbb{R}^3$ on the visual cortical surface, pRF is used to determine its receptive field, including its center location $v$ and size $\sigma$ in the visual field. Namely, the collection of the mapping between $P$ and $(v,\sigma)$ constitutes a raw retinotopic map. Assuming that the voxel's population receptive field model is $r(v';v,\sigma)$ and the hemodynamic function is $h(t)$, the predicted fMRI signal of the voxel can be written as:

$$\hat{y}(v, \sigma, n) = \beta \left( \int r(v'; v, \sigma) s(t, v') dv' \right)^n * h(t), \tag{1}$$

where $\beta$ is the activation level, $n$ is the exponent of the power function. We used the standard population receptive field model, i.e., $r(v'; v, \sigma) = \exp\left(-\frac{(x'-x)^2+(y'-y)^2}{2\sigma^2}\right)$, where $x = v^{(1)} \cos v^{(2)}, y = v^{(1)} \sin v^{(2)}$ (similarly for $x', y'$). The center of the receptive field $\boldsymbol{v}$ and population receptive field size $\sigma$ can be estimated by minimizing the Least Square difference between the measured and predicted fMRI activation levels:

$$(\boldsymbol{v}, \sigma, n) = \arg \min_{(v,\sigma,n)} |\hat{y}(\boldsymbol{v}, \sigma) - y(P)|^2, \tag{2}$$

where $y(P)$ is the fMRI signal at voxel $P$. Iterations of this procedure across all the voxels on the visual cortical surface generate the raw retinotopic map. The quality of fit of the pRF model for voxel can be evaluated by $R^2 = 100(1 - \int (\hat{y} - y)^2 dt / \int y^2 dt)$.

## Topological smoothing

**Quasiconformal mapping and Beltrami coefficient.** Most surface mappings can be modeled by quasi-conformal mapping, a generalization of conformal mapping. Specifically, the local deformation of a quasiconformal map can be characterized by its associated Beltrami coefficient [39]. Mathematically, $f:\mathbb{C}\to\mathbb{C}$ is quasiconformal if it satisfies the Beltrami equation,

$$\frac{\partial f}{\partial \bar{u}} = \mu_f \frac{\partial f}{\partial u} \tag{3}$$

for some complex-valued Lebesgue measurable function $\mu_f$ satisfying $\|\mu_f\|_\infty < 1$. The complex-valued $\mu_f = \frac{\partial f}{\partial \bar{u}} / \frac{\partial f}{\partial u}$ is called the Beltrami coefficient of $f$, with $u = u^{(1)} + iu^{(2)}$ and $\bar{u} = u^{(1)} - iu^{(2)}$.

Intuitively speaking, if $\|\mu_f\|_\infty < 1$, the map preserves the orientations of all triangles and therefore preserves the topological condition. However, if $\|\mu_f\|_\infty > 1$ the topological condition is violated because the orientations of some triangles are flipped.

**Surface flattening.** Let $f_r: v_i \mapsto P_i$ be the retinotopic map between the retina and V1 cortical surface. To use the tool of the Beltrami coefficient, we need to flatten the cortical surface from 3D to a 2D unit disk. Because topological proprieties are transitive (that is if mappings $f_r$ and $c$ are both topological, the composed mapping $f_r \circ c$ is also topological) and mutual (if mappings $f_r$ is topological, $f_r^{-1}$ is also topological), we can reformulate the topological condition by mapping the 3-dimensional cortical surface to a 2D parametric space with a topological transformation [37,40]. Here, we performed a conformal mapping (angle preserving) of the visual cortical surface to a parametric space $c: P_i \mapsto \boldsymbol{u_i}$, where $\boldsymbol{u_i} = (u_i^{(1)}, u_i^{(2)}) \in \mathbb{R}^2$. Although it is angle preserving and does not introduce any angle distortions, conformal mapping may introduce big metric distortions. To reduce metric distortions, we cut a geodesic disk patch that contains the region of interest from the cortical surface by: (1) picking a point on the cortical surface that roughly corresponds to the fovea as the center of the disk, (2) calculating the geodesic distances from all points on the cortical surface to the center of the disk [41], and (3) keeping the points of the cortical surface whose geodesic distances to the center of the disk are within a certain value. The geodesic disk patch (the region-of-interest—*ROI*—shown in gray color in Fig 4B) is mapped to a unit disk (Fig 4C). We refer the readers to [31,42] for conformal mapping and [43] for the computation of geodesic distance.

In this work, we focus on the 2D retinotopic mapping $f = f_r^{-1} \circ c^{-1}$ instead of the 3D retinotopic mapping $f_r$.

**Mathematical model.** With the conformal parametrization in the last section, we use the Beltrami coefficient $\mu_f$ to model topological smoothing by finding new visual coordinates $\hat{\boldsymbol{v}} \equiv$

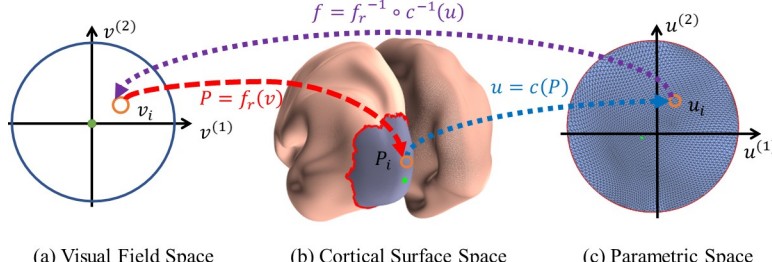

(a) Visual Field Space        (b) Cortical Surface Space        (c) Parametric Space

**Fig 4.** Illustration of (A) the visual field space, (B) the cortical surface space, and (C) the parametric space.

$\hat{f}$ such that,

$$\hat{\boldsymbol{v}} = \hat{f} = \arg\ \min_{\hat{\boldsymbol{v}}} \int w|\hat{f} - \boldsymbol{v}|^2 + s|\nabla\hat{f}|^2 \mathrm{ds},\ \text{s.t.} \|\mu_{\hat{f}}\|_\infty < 1. \tag{4}$$

where $w$ is the weight function, $\nabla = (\partial/\partial u^{(1)}, \partial/\partial u^{(2)})$ is the gradient operator, $s$ is a positive scalar emphasizing the smoothness and $\mu_{\hat{f}}$ is the Beltrami coefficient of $\hat{f}$ written as (see *S2 Text*):

$$\mu_{\hat{f}} = \left(\frac{\partial\hat{f}}{\partial u^{(1)}} + i\frac{\partial\hat{f}}{\partial u^{(2)}}\right) \Big/ \left(\frac{\partial\hat{f}}{\partial u^{(1)}} - i\frac{\partial\hat{f}}{\partial u^{(2)}}\right), i = \sqrt{-1}. \tag{5}$$

In Eq 5, we have rewritten a point in the 2D parametric domain as a complex number $u = u^{(1)} + iu^{(2)}$ and the corresponding visual coordinate as $\hat{f} = \hat{f}^{(1)} + i\hat{f}^{(2)}$. Because we will only discuss the problem in 2D in the subsequent sections of the paper, we interchangeably regard each complex number as a position vector or vice versa, that is, $u = u^{(1)} + iu^{(2)}$ is regarded as a complex number or as a position vector $\boldsymbol{u} = (u^{(1)}, u^{(2)})$. We denote a position vector with a bold $\boldsymbol{u}$ and a complex number with an italic $u$. The same notation applies to $\boldsymbol{v}$.

The Beltrami of $\hat{f}$ can be used to quantify the Beltrami of $f_r^{-1}$ because the Beltrami coefficient does not change with the conformal mapping, that is, $\mu_f = \mu_{f_r^{-1}}$ if $c$ is a conformal mapping (see *S2 Text* for mathematical details). Intuitively, the Beltrami coefficient measures the angle distortion in a mapping (see *S4 Text*), incorporating a conformal mapping $c$ does not introduce any angle distortion and therefore does not change the original Beltrami coefficient. We can directly estimate the Beltrami coefficient of the retinotopic mapping in the 2D parametric domain on the unit disk based on this property.

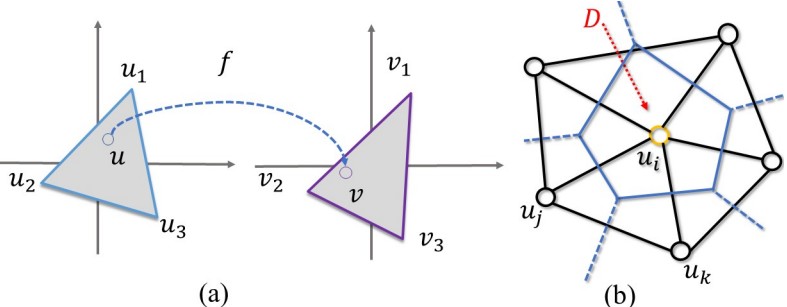

(a)        (b)

**Fig 5. Illustration of a mapping function and the divergence computation.** (A) An illustration of the mapping function in the discrete domain, and (B) the divergence approximation for a vertex.

**Model solver.**   To solve Eq (4) efficiently, we divide the problem into two subproblems, namely smoothing and topological projection. Specifically, during smoothing, we ignore the topological condition and find a smoothed map by,

$$\tilde{\boldsymbol{v}} = \arg \min_{\tilde{v}} \int w|\tilde{\boldsymbol{v}} - \boldsymbol{v}|^2 + s|\nabla\tilde{\boldsymbol{v}}|^2 \mathrm{d}s \qquad (6)$$

which can be solved by Laplacian smoothing [44,45] efficiently. For the second subproblem, we ignore the smoothness term and try to make the results topological.

$$\hat{\boldsymbol{v}} = \arg \min_{\hat{v}} \int w|\hat{\boldsymbol{v}} - \tilde{\boldsymbol{v}}|^2 \mathrm{d}s, \mathrm{s.t.} \|\mu_{\hat{v}}\|_{\infty} < 1 \qquad (7)$$

Since we require the results to be both topological and smooth, we solve the subproblems iteratively. Namely, the output of topological projection is fed to smoothing, and so on so forth.

The key challenge is to solve Eq (7) efficiently. Because the topological condition is formulated with the Beltrami coefficient ($\|\mu_{\hat{v}}\|_{\infty} < 1$), we need tools to compute the Beltrami coefficient and recover the function after processing the Beltrami. In the following, we describe the Beltrami coefficient computation when a function is given and the function recovery when the Beltrami coefficients are given in the discrete setting. Then we elaborate on the details about solving each subproblem.

**Model discretization.**   We now convert the problem to the discrete domain because a cortical surface, $S$, is typically described by vertices $V_S$ and triangular faces $F_S$: $S = (F_S, V_S)$. The visual field is also discretized. Retinotopic mapping can be reformulated as finding the corresponding visual coordinate $v_i$ in the visual field for every vertex $P_i \in V_S$. We denote $W_i$ as $w(v_i)$ to emphasize/deemphasize voxels with high and low quality of pRF fits.

$$E = \sum_i W_i|v_i - \hat{f}_i|^2 + s|\nabla\hat{f}_i|^2 + \lambda|\mu_i|^2, \mathrm{s.t.}|\mu_{\hat{f}_i}| < 1. \qquad (8)$$

$\hat{f}_i = \hat{f}(u_i)$ is the coordinate to be solved, $\mu_i = \mu_{\hat{f}}(u_i)$ is the Beltrami coefficient at point $u_i$.

**Discrete Beltrami coefficient computation.**   Given the analytical form of a mapping function $f$, we can directly compute the complex-valued Beltrami coefficient $\mu_f$ according to Eq (5). However, in the discrete case, the function is only defined on vertices, i.e., we only know the mapping function $f$ that maps the vertices on the unit disk to the vertices on the visual field (Fig 5A), that is, $v_1 = f(u_1)$, $v_2 = f(u_2)$, and $v_3 = f(u_3)$. To approximate the derivatives, $f$ is interpolated linearly on each triangle, i.e., for any point $u$ within the triangle, $f(u) = B_1(\boldsymbol{u})v_1 + B_2(\boldsymbol{u})v_2 + B_3(\boldsymbol{u})v_3$. The coefficients, $B_1$, $B_2$, and $B_3$ are called the barycentric coefficients. Specifically, $B_1$ (similarly for $B_2$ and $B_3$) is the ratio between the areas of triangles $\Delta uu_2u_3$ and $\Delta u_1u_2u_3$: $B_1(u) = \mathrm{Area}(\Delta uu_2u_3)/\mathrm{Area}(\Delta u_1u_2u_3)$. Now we can compute $f$'s partial derivatives to $u^{(1)}$ and $u^{(2)}$ and therefore the Beltrami coefficient $\mu_f$ according to Eq (5) (see *S3 Text* for the explicit expression). Clearly $\mu_f$ is a constant within each triangle, as we approximate the mapping function linearly within each triangle.

**Recovering mapping with Beltrami coefficients.**   We now explain how to recover function $f = f^{(1)} + if^{(2)}$ with the given Beltrami coefficient function $\mu = \rho(u) + i\tau(u)$, first introduced in [32]. According to the definition in Eq (5), we have:

$$\left(\frac{\partial f}{\partial u^{(1)}} + i\frac{\partial f}{\partial u^{(2)}}\right) / \left(\frac{\partial f}{\partial u^{(1)}} - i\frac{\partial f}{\partial u^{(2)}}\right) = \rho + i\tau. \qquad (9)$$

Based on its real and imaginary components, Eq (9) is equivalent to:

$$
\begin{cases}
-\dfrac{\partial f^{(2)}}{\partial u^{(2)}} = \alpha_1 \dfrac{\partial f^{(1)}}{\partial u^{(1)}} + \alpha_2 \dfrac{\partial f^{(1)}}{\partial u^{(2)}} \\[2mm]
\dfrac{\partial f^{(2)}}{\partial u^{(1)}} = \alpha_2 \dfrac{\partial f^{(1)}}{\partial u^{(1)}} + \alpha_3 \dfrac{\partial f^{(1)}}{\partial u^{(2)}}
\end{cases},
\tag{10A)(10B}
$$

where $\alpha_1 = \frac{(\rho-1)^2 + \tau^2}{\rho^2 + \tau^2 - 1}$, $\alpha_2 = \frac{-2\tau}{\rho^2 + \tau^2 - 1}$, and $\alpha_3 = \frac{(\rho+1)^2 + \tau^2}{\rho^2 + \tau^2 - 1}$. Applying $\frac{\partial}{\partial u^{(1)}}$ on Eq (10A) and $\frac{\partial}{\partial u^{(2)}}$ on Eq (10B), and then adding them together, we can simplify Eq (9) as:

$$
\nabla \cdot A \nabla f^{(1)} = 0,
\tag{11}
$$

where $A = \begin{pmatrix} \alpha_1 & \alpha_2 \\ \alpha_2 & \alpha_3 \end{pmatrix}$, $\nabla f^{(1)} = (\partial f_c^{(1)}/\partial u^{(1)}, \partial f_c^{(1)}/\partial u^{(2)})$ is the gradient of $f^{(1)}$, and $\nabla$ is the divergence operator ($\nabla \cdot \boldsymbol{G} = \partial G^{(1)}/\partial u^{(1)} + \partial G^{(2)}/\partial u^{(2)}$ for vector $\boldsymbol{G} = (G^{(1)}, G^{(2)}) = A\nabla f^{(1)}$). By solving Eq (11) with Dirichlet boundary conditions (i.e., $f^{(1)}$ values on the boundary of the cortical patch), we can find a unique solution of $f^{(1)}$. Similarly, $f^{(2)}$ can be written as $\nabla \cdot A \nabla f^{(2)} = 0$ and be solved uniquely.

In the discrete case, the mapping function is interpolated on each triangle. The gradient operator $\nabla f^{(1)}(u)$ can be written out and is a constant vector on each triangle. The divergence $\nabla \cdot \boldsymbol{G}$ is approximated on the dual of each vertex (a convex cell consisted of the circumcenters of the neighbor triangles). Specifically, consider a vertex with its neighbors (Fig 5B). We assume that $\nabla \cdot \boldsymbol{G}$ is a constant on the dual of the center vertex, i.e., polygon D, enclosed by circumcenters of the neighbor triangles. Namely, we approximate the term with $\nabla \cdot \boldsymbol{G}(u_i) = \frac{1}{|D|} \int_D \nabla \cdot \boldsymbol{G} d\sigma$. According to the divergence theorem [46], the divergence $\int_D \nabla \cdot \boldsymbol{G} d\sigma$ can be written as the integral on the boundary, i.e. $\nabla \cdot \boldsymbol{G}(u_i) = \frac{1}{|D|} \int_D \nabla \cdot \boldsymbol{G} d\sigma = \frac{1}{|D|} \int_{\partial D} \boldsymbol{G} \cdot (\boldsymbol{n} \times d\boldsymbol{l}) ds$, where $\boldsymbol{n}$ is the dual cell unit normal. Since $\boldsymbol{G}$ is a constant vector on each triangle (in the discrete case), the integration can be written as:

$$
\nabla \cdot \boldsymbol{G}(u_i) = \frac{1}{|D|} \int_{\partial D} \boldsymbol{G} \cdot (\boldsymbol{n} \times d\boldsymbol{l}) = \frac{1}{|D|} \sum_{[u_i, u_j, u_k] \in N(u_i)} \boldsymbol{G}_{T_l} \cdot \frac{\boldsymbol{s}_i}{2},
\tag{12}
$$

where $\boldsymbol{G}_{T_j}$ is the value of $\boldsymbol{G}$ within triangle $T_j$, and $\boldsymbol{s}_i = (u_k - u_j)^{\perp}$ denotes a vector perpendicular to $u_k - u_j$ with the same norm as $u_k - u_j$. Substituting Eq (12) into Eq (11), we have a set of linear equations for each $f^{(1)}(u_i)$ and its neighbors. Finally, we can write $\nabla \cdot A \nabla f^{(1)} = 0$ in a matrix form (see *S3 Text* for the explicit form) and solve $f^{(1)}$ efficiently with the Dirichlet boundary condition. Similarly, $f^{(2)}$ can be solved efficiently.

Next, we describe how we can optimize Eq (7) by solving two subproblems iteratively.

## Solving the first subproblem: Laplacian smoothing

The first subproblem can be solved by *Laplacian smoothing* of $f^{(1)}$ and $f^{(2)}$ separately. To get a smooth scalar $\hat{f}^{(1)}$ from a noisy $f^{(1)}$, we solve the following Laplacian smoothing equation [44,45]:

$$
\hat{f}^{(1)} = \arg \min \int |\nabla \hat{f}^{(1)}|^2 + s|f^{(1)} - \hat{f}^{(1)}|^2 du,
\tag{13}
$$

where the $\nabla$ is defined in the parametric domain, i.e., $\nabla = (\partial/\partial u^{(1)}, \partial/\partial u^{(2)})$. To find the solution, we set the partial derivatives of Eq (13) to zero. It leads to the following equation $(-\nabla \cdot \nabla + 2s\,I)\hat{f}^{(1)} = 2sf^{(1)}$. It can be solved efficiently with linear algebra because $\nabla^2 = \nabla \cdot \nabla$ can be written in a matrix form (the details are provided in *S3 Text*), and $\sigma$ is optimally chosen

to minimize the generalized cross-validation score [47]. Similarly, $f^{(2)}$ can be smoothed to $\hat{f}^{(2)}$. Since the Beltrami coefficients quantify local deformations, Laplacian smoothing on visual coordinates will smooth the Beltrami coefficients. If most triangles are correctly orientated, the smoothing will also reduce the number of flipped triangles.

## Solving the second subproblem: Topological projection

The second subproblem is to make $f$ topological. It means that the associated Beltrami coefficients must satisfy the condition $\|\mu_f\|_\infty < 1$ on the entire cortical surface. To ensure topological condition, for any mapping whose $|\mu|$ is greater than 1, we scale its magnitude by $\mu' = \mu/(|\mu| + \epsilon)$. The procedure is called "topological projection" [48,49] or "chopping" [50]. The goal is to find a topological mapping that is close to the original non-topological mapping. The new $\mu'$ shares the same argument of $\mu$, and $|\mu'|$ is less than 1 when we choose a small positive $\epsilon$. Because the Beltrami coefficient uniquely encodes the mapping [31,51], the projection process changes the mapping when we solve equation Eq (11). The process of making the mapping topological can be illustrated with the example in Fig 2B, in which the target $f_i$ is below the horizontal axis, and the magnitude of the associated Beltrami coefficient is greater than one ($|\mu_{\Delta f_i f_j f_k}| > 1$). If we shrink the Beltrami coefficient for $\Delta f_i f_j f_k$ and retain the Beltrami coefficients for all the other triangles, the new Beltrami coefficient $|\mu'_{\Delta f_i f_j f_k}|$ is less than 1. When we solve the mapping function according to the new Beltrami coefficient (Eq (11)), $f_i$ is moved and placed above the horizontal axis, making the triangle $\Delta f_i f_j f_k$ topological. Note that other target points, $f_l$, $f_m$, and $f_n$, also move to maintain the Beltrami coefficients of their associated triangles, e.g., when $f_i$ moves up, $f_l$ moves accordingly so that the Beltrami coefficient of triangle $\Delta f_i f_k f_l$ stays the same. In summary, given an initial non-topological mapping, we can project the Beltrami coefficient and solve Eq (11) to obtain a topological mapping that is close to it.

**Removing phase jumping.** We adopt the widely used polar coordinate system for the visual field, i.e., eccentricity $r = v^{(1)}$ and polar angle $\theta = v^{(2)} \in [0,2\pi]$. The polar coordinate system has many advantages, but the polar angle $\theta$ is not continuous numerically near the $x$ axis even when the mapping itself is. As shown in Fig 6A, points $v_1$ and $v_2$ share the same eccentricity and are both very close to the $x$ axis, but their polar angles are very different because the polar angle jumps from $2\pi$ to $0$ near the positive $x$ axis. We call the phenomena "phase jumping". It makes the coordinates around the $x$ axis discontinuous.

Phase jumping is not an intrinsic property of retinotopic mapping. The issue can be addressed with a transformed coordinate system: If we rotate the polar coordinate system, the phase jumping axis can be shifted. To avoid phase jumping, we shift the zero polar angle to the negative $x$ axis for the left hemisphere (Fig 6B). Now phase jumping occurs near the negative $x$ axis. Because retinotopic mapping of the left hemisphere is in Quadrants *I* and *IV*, the shift

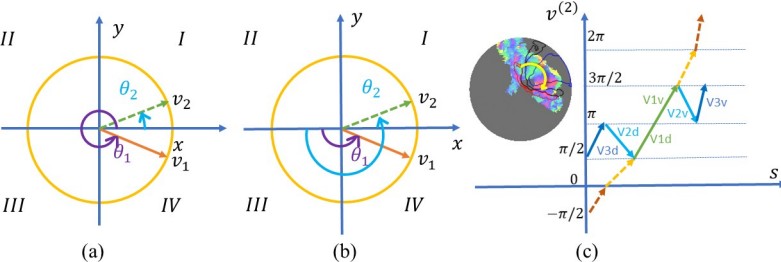

(a)     (b)     (c)

**Fig 6. Removing phase jumping and phase reversal.** (A, B) Phase jumping and the shift scheme used to remove it. (C) Phase reversal and the transformation used to remove it.

**Table 1. Polar angle transformations are used to remove phase-jumping and changes of visual field sign for the left and right hemispheres in multiple visual areas.**

| Region Name | Transformation of Polar Angle $\theta$ | |
| --- | --- | --- |
| | Left Hemisphere | Right Hemisphere |
| V1v | $T(\theta) = \theta+\pi$ | $T(\theta) = \theta$ |
| V1d | $T(\theta) = \theta-\pi$ | $T(\theta) = \theta$ |
| V2v | $T(\theta) = -\theta+2\pi$ | $T(\theta) = -\theta+\pi$ |
| V2d | $T(\theta) = -\theta-2\pi$ | $T(\theta) = -\theta-\pi$ |
| V3v | $T(\theta) = \theta+2\pi$ | $T(\theta) = \theta+\pi$ |
| V3d | $T(\theta) = \theta-2\pi$ | $T(\theta) = \theta-\pi$ |

allows us to avoid phase jumping. For the right hemisphere, retinotopic mapping is in Quadrants *II* and *III*; we do not need any shift. By doing this, we can avoid unnecessary visual coordinate discontinuities.

**Removing phase reversal.** We apply another linear transformation on the polar angle to remove phase reversal across visual areas. Phase reversal is an important intrinsic property of retinotopic mapping. As shown in Fig 6C, if one walks from V3d to V3v along the yellow arc, the corresponding polar coordinate $v^{(2)}$ on the visual field first increases and then decreases. We assign a visual area with a positive (negative) visual field sign if the polar coordinate increases (decreases) when we move clockwise on an equal eccentricity curve, e.g., the yellow arc in Fig 6C. A phase reversal across the boundary of two cortical areas causes a change of visual field sign and therefore breaks the topological condition for the points near the boundary between visual areas. To make our proposed smoothing method applicable to multiple visual areas simultaneously, we apply another linear transformation to remove the visual field sign changes across visual area boundaries. The combination of two transformations, one for removing phase jumping and the other for eliminating visual field sign changes, is summarized in Table 1. We call the transformed polar angle "*extended polar angle*". Because polar angles and visual area labels have a one-to-one correspondence with the extended polar angles, one can recover the original polar angles and visual area labels from the extended polar angles. With accurate extended polar angles, one can draw accurate interior boundaries between different visual areas (V1 and V2, V2 and V3, etc).

In addition, these transformations make the smoothing more precise, especially near the boundaries of visual areas, because they remove the sharp polar angle changes across the boundaries.

**Algorithm.** We now summarize the overall topological smoothing algorithm in Alg. 1. The input data consist of the conformal parameterization $u$ of the cortical surface, and their associated visual field coordinates $v$. The output is the updated retinotopic map coordinate $\hat{v} = \hat{f}$. The algorithm is implemented in MATLAB 2019b [52,53].

**Algorithm 1.** Topological Smoother
**Input Data:** Retinotopic coordinates $v = \{v_i\}$, conformal parametrization $u = \{u_i\}$ of the cortical surface, initial boundary retinotopic coordinates, $v_B$, and boundary value change tolerance $\epsilon$.
**Output:** Smoothed retinotopic coordinates $\hat{v} = \{\hat{v}_i\}$
Correct the visual coordinates according to Table 1 to get adjusted visual coordinates $v'$
$\hat{v}_i \leftarrow v'_i$, assign the initial retinotopic coordinates for each vertex on the unit disk
**repeat**
    Compute Beltrami coefficient $\mu$ for the mapping $\hat{f} : u \rightarrow \hat{v}$ using Eq (5).

```
     Project non-topological mappings with μ' = μ/(|μ|+ε), if |μ|>1.
     Compute new mapping v̂' by LBS for μ' with boundary value v_B.
     Apply Laplacian smoothing on v̂'(v̂'^(1) and v̂'^(2), respectively) to get v̂.
     Fit a linear function near the boundary and update the boundary
value v_B to v'_B
     Check and restore each point in v'_B if it deviates from the initial
value by more than ε.
until max|μ|<1
return v̂.
```

## Results

### Synthetic data

We first evaluated the performance of the algorithm on synthetic data generated with the following function,

$$u^{(1)} + iu^{(2)} = k\ln(v^{(1)} + iv^{(2)}). \tag{14}$$

We chose ln function ($k = 0.5$ in this paper) as it is a good approximation of the retinotopic mapping from the visual field to the flattened cortical surface [54]. Note that in the original study, Schwartz used the model to map from the visual field to the parametric coordinates of a flattened cortical surface. It is the inverse function that we studied in this paper. We followed the convention only in this subsection, i.e., generating a smoothed $\hat{u}(v)$ from a noisy $u(v)$, instead of a smoothed $\hat{v}(u)$ from a noisy $v(u)$. Note that, because the topological condition is reciprocal, the conclusion based on this approach still applies to the inverse function. More specifically, the visual field grid mesh is first created (Fig 7A) with the following configurations: eccentricity spans from 0° to 4.5° and polar angle ranges from −π/2 to π/2. We then sampled 12 points in each direction and formed the quadrilateral grid mesh (144 points in total). Based on the quadrilateral mesh, we created the triangular faces by connecting diagonal points for each quadrilateral. Then we mapped the triangular mesh to the parametric space (Fig 7B), according to Eq (14). Ideally, the mapping points should be located on the grid (Fig 7B). We then added a small amount of noise with a peak signal-noise ratio (PSNR) of 10 (Fig 7C), and a large amount of noise with a PSNR = 5 (Fig 7D).

We applied several smoothing methods, including Average Smoothing, Median Smoothing, and Laplacian Smoothing [44], as well as our proposed Topological Smoothing, to the synthetic mapping data in both high and low signal-to-noise cases. All the existing methods are spatial smoothing procedures. Intuitively, Average Smoothing takes the average around the neighborhoods of each vertex. Median Smoothing takes the median value of the

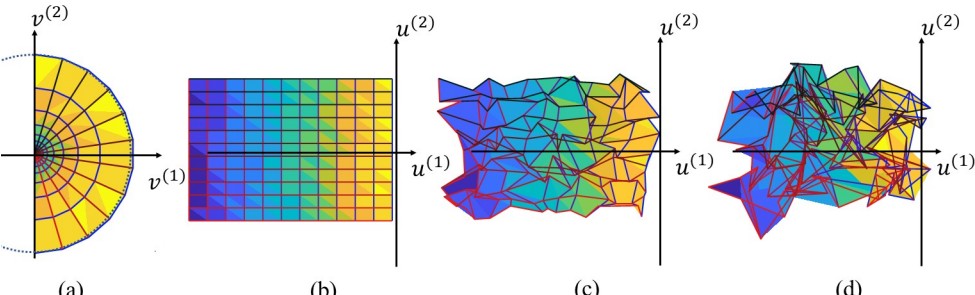

(a)                    (b)                    (c)                    (d)

**Fig 7. Illustration of the synthetic data.** (A) The visual field domain. (B) Mapping without noise. (C) Mapping with a small amount of noise (PSNR = 10). (D) Mapping with a large amount of noise (PSNR = 5).

**Table 2. Comparison of different smoothing methods based on three metrics: value deviation, angle distortion, and number of flipped triangles with a small amount of noise (PSNR = 10).**

| Method | Value Distortion | | | Angle Distortion in deg. | | | Topology | |
|---|---|---|---|---|---|---|---|---|
| | **Mean** | **Std** | ***p*** | **Mean** | **Std** | ***p*** | $\hat{T}$ | ***p*** |
| No Smoothing | 0.235 | 0.13 | 0.00001 | 45.001 | 25.25 | 0.00001 | 54 | 0.00000 |
| Average | **0.134** | **0.09** | 0.00001 | 30.903 | 22.52 | 0.00005 | 10 | 0.00000 |
| Median | 0.170 | 0.10 | 0.00001 | 40.124 | 24.04 | 0.00001 | 19 | 0.00000 |
| Laplacian | 0.558 | 0.48 | 0.00001 | 41.432 | 25.16 | 0.00001 | 18 | 0.00000 |
| Proposed | 0.143 | **0.09** | ------ | **18.313** | **16.41** | ——— | **0** | ——— |

neighborhood. All existing smoothing procedures treat the $u^{(1)}$ and $u^{(2)}$ coordinates separately. In this section, we fixed the smooth parameter $s = 0.001$ (Eq (7)) and the boundary values were initially inferred from the Average Smoothing results and boundary change tolerance is $\epsilon = 0.01$. We list the performance metric, i.e., the deviation from the ground truth of all smoothers in Tables 2 and 3. Value deviation is the average difference relative to the ground truth. Angle distortion (the angle spanned by the tangent vector of $u^{(1)}$'s contour curves and the tangent vector of $u^{(2)}$'s contour curves, at the same location; see *S4 Text* for details) quantifies the local angle changes (in degree unit) after the mapping, and the number of flipped triangles $\hat{T}$ measures violations of the topological condition. All the experiments were repeated 50 times to reduce fluctuation caused by random noise. The number of flipped triangles $\hat{T}$ is the median of the 50 experiments. In addition, we ran a non-parametric permutation test [55] to conduct pairwise comparisons of the results from the proposed method and each of the other methods (including no smoothing). The null hypothesis is that the results from the proposed smoother are not significantly different from those of the other methods. We report the *p* value, i.e., the probability that the null hypothesis was true. In brief, if the *p* value is less than 0.01, the results from the proposed smoothing method are statistically different from those of the other methods.

We found that (1) all smoothing methods reduced value deviation, angle distortion, and number of flipped triangles (Tables 2 and 3); (2) the proposed method achieved the best topological result and the smallest angle deviation (all p<0.00001); (3) when the noise was relatively small (Table 2), violations of the topological condition were largely corrected by most of the methods, and the average smoothing method achieved very good topological results; (4) when the noise was relatively high (Table 3), the average smoothing method lost its ability to correct topological violations; and (5) the proposed method always performed best in angle preserving, which was not a surprise as the magnitude of Beltrami coefficient |μ| is related to angle distortion of the mapping (see *S4 Text*), and we penalized big |μ|'*s* in the proposed method (Eq

**Table 3. Comparison of different smoothing methods based on three metrics: value deviation, angle distortion, and number of flipped triangles with a large amount of noise (PSNR = 5).**

| Method | Value Distortion | | | Angle Distortion (degree) | | | Topology | |
|---|---|---|---|---|---|---|---|---|
| | **Mean** | **Std** | ***p*** | **Mean** | **Std** | ***p*** | $\hat{T}$ | ***p*** |
| No Smoothing | 0.334 | 0.18 | 0.00001 | 47.067 | 25.87 | 0.00001 | 65 | 0.00001 |
| Average | **0.169** | 0.10 | 0.00001 | 36.099 | 24.09 | 0.00040 | 22 | 0.00001 |
| Median | 0.206 | 0.12 | 0.00001 | 43.314 | 24.76 | 0.00001 | 32 | 0.00001 |
| Laplacian | 0.604 | 0.46 | 0.00001 | 44.544 | 26.18 | 0.00001 | 25 | 0.00001 |
| Proposed | **0.169** | **0.11** | ------ | **23.226** | **20.70** | ------ | **0** | ——— |

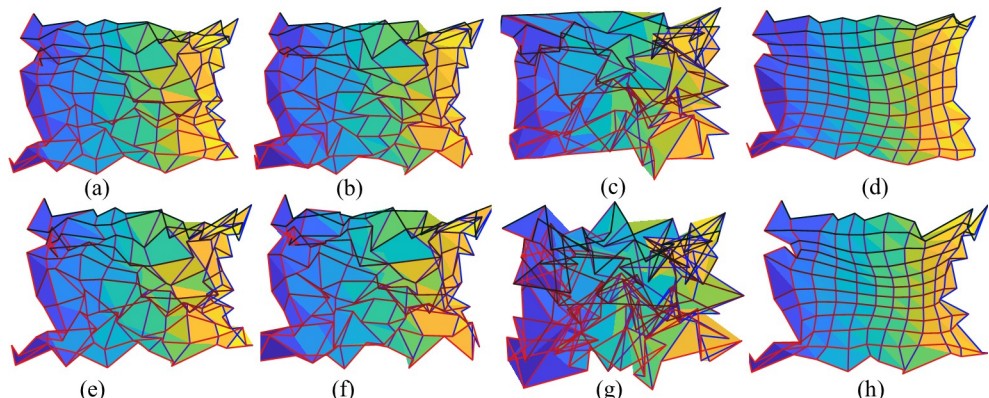

**Fig 8.** Smoothing results for the synthetic data: (A)-(D) are the results with PSNR = 10 for (A) Average smoothing, (B) Median smoothing, (C) Laplacian smoothing, and (D) proposed topological smoothing; (E)-(H) are the results with the same order as (A)-(D) but with PSNR = 5.

(7)) to reduce angle distortions. To summarize, the proposed smoother performed best to correct topological violations and reduce angle distortions. It also performed better or comparably with all the other methods in terms of visual coordinate distortions. Fig 8 illustrates the smoothing results for all the methods when the noise was relatively low (PSNR = 10).

## Synthetic Data II

We performed another set of synthetic data experiment in which noise was directly added to the fMRI signals. We started with the ground-truth parameters: receptive field centers spanning the right visual space [0:0.08:6.4]˚×[0,0.08: 4.8]˚; receptive field size = 0.4˚; gain = 1; and exponent of the power function $n$ = 0.5. Then, we followed Kay's pRF model (AnalyzePRF [38,56] with stimulus from the HCP group, 200×200 in spatial resolution with 1800 seconds) to generate the ideal fMRI signal. We further normalized the fMRI signal to unit variance, added a specific amount of white noise, decoded the pRF model for the fMRI signal (with noise), and computed the number of flipped triangles. We varied the noise level until the percentage of flipped triangle was about 20%. (Specifically, in our experiments, noise with a 2.5 standard deviation induced about 22% flipped triangles). With pRF decoding results at this noise level as input, we applied the topological smoothing method and generated new visual coordinates. Then we compared our work with the pRF model and several post-processing methods (Table 4).

   We found that (1) the proposed method can fully ensure the topological condition, with 0 flipped triangles, while other smoothing methods had at least 6% of flipped triangles; (2) the proposed method had minimal angle distortion; and (3) all smoothing methods improved the receptive field center estimates, with the median smoothing method achieving minimal center

**Table 4. Comparison of different smoothing methods based on noisy fMRI decoding with about 22% of flipping triangles.**

| Method | Center Distortion | | | Angle Distortion (degree) | | | $\hat{T}$ | $R^2$ |
|---|---|---|---|---|---|---|---|---|
| | Mean | Std | p | Mean | Std | p | | |
| No Smoothing | 0.407 | 0.520 | 0.352 | 48.55 | 30.20 | 0.00 | 32 | 13.28 |
| Average | 0.311 | 0.238 | 0.070 | 23.35 | 23.16 | 0.00 | 15 | 13.41 |
| Median | 0.298 | 0.176 | 0.010 | 22.40 | 16.57 | 0.00 | 11 | 13.41 |
| Laplacian | 0.429 | 0.509 | 0.149 | 42.82 | 29.48 | 0.00 | 23 | 13.09 |
| Proposed | 0.361 | 0.231 | ------ | 13.84 | 12.87 | ------ | **0** | 13.27 |

distortion. The last point is reasonable because the current work is mainly focused on the topological condition. Future fine-tuning of the pRF parameters may alleviate this problem. The new synthetic experimental results further demonstrated that the proposed method could remove topological violations and reduce angle distortions.

## V1 retinotopic map

Although 7-Tesla MRI systems have dramatically improved the signal-to-noise ratio, the retinotopic maps in the HCP dataset are still quite noisy and non-topological. We first applied our proposed smoother on the V1 retinotopic maps of all observers in the HCP dataset.

Since there is no ground truth, we evaluated the performance of the method by the goodness of fit of the predicted fMRI signals with the smoothed visual coordinates. Specifically, we first decoded the visual coordinates $v$, population perception field size $\sigma$, BOLD response level $g$ for the 181 observers by the pRF method [3]. Then we applied several smoothing methods on their V1 retinotopic maps. Based on the smoothed coordinates, we computed fMRI time series predictions at every vertex of the cortical surface. To evaluate the impact of changing the receptive field centers, we set the response level $g$ and the exponent of the power function $n$ the same as the original pRF in computing fMRI time series predictions.

We measured the goodness of fit of the smoothing methods to the fMRI time series (with the code from AnalyzePRF [38,56]). The results are listed in Table 5, with the following metrics: (1) the average number of flipped triangles across subjects, (2) the average percentage of the non-topological area $Ar_n$, (3) the average Normalized-Root-Mean-Square-Error (NRMSE) of the fits, and (3) the average variance $\bar{R}^2$ accounted for across all the vertices in V1.

We observe that: (1) all the smoothing methods reduced the number of flipped triangles, while only the proposed method eliminated topology violations completely; (2) the percentage of the non-topological area $Ar_n$ was about 8~20% of the total area with all but the proposed smoothing method; (3) The proposed method decreased $R^2$, mainly due to the constrain of the topological condition. The number of flipped triangles for individual observers by all methods can be found in the supplementary data (see *Table A in S5 Text*). The results also show that the proposed method was always better in preserving topology while the goodness of fit to the fMRI time series was reduced from the original result.

For a more intuitive comparison, Fig 9 shows the original retinotopic map of the left hemisphere of the first observer in the HCP dataset and the smoothed versions by different methods. The original retinotopic map (Fig 9A) is not topological because there are many flipped triangles. Fig 9B–9E show the results from the Average, Median, Laplacian, and proposed smoothing methods, respectively. Although all smoothing methods improved the smoothness of the maps, topological violation still occurred in the results of all but the proposed smoothing methods, especially near the fovea. Only our proposed method can generate topology-preserving results (Fig 9E). Fig 9F shows the smoothed retinotopic map with level sets on the inflated left hemisphere. To enable a clear comparison, we present enlarged level sets on the inflated

**Table 5. Comparison of different methods in fitting the fMRI time series.**

| Method | $\hat{T}$ | $Ar_n$(%) | $NRMSE \times 10^{-3}$ | $\bar{R}^2$ |
|---|---|---|---|---|
| Before Smoothing | 73 | 20.7 | **0.07757** | **28.23** |
| Average Smoothing | 28 | 8.1 | 0.07779 | 27.86 |
| Median Smoothing | 35 | 9.7 | 0.07773 | 27.92 |
| Laplacian Smoothing | 64 | 18.0 | 0.07763 | 28.09 |
| Proposed Smoothing | **0** | **0.0** | 0.08139 | 20.35 |

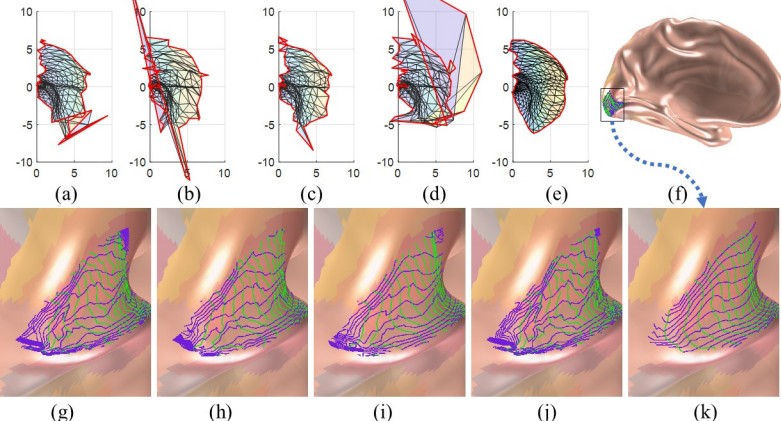

**Fig 9.** Results on the first observer: (A) the raw retinotopic map on the 2D parametric domain; (B)-(E) results from the Average, Median, Laplacian, and proposed methods, respectively; (F) the whole inflated left mesh; (G) the raw map on the cortical surface; (H)-(K) results on the inflated surface, in the same order of (B)-(E). In (A)-(E) the $x$ axis is the visual $x$ coordinate, and the $y$ axis is the visual $y$ coordinate.

mesh for each smoothing method in Fig 9H–9K. The green curves from the left to right represent 0.5˚ to 6˚ eccentricities with a 0.5˚ interval, and the blue curves, from the bottom to up, represent -90˚ to 90˚ polar angles with a 15˚ interval. We can also clearly see topological violations in Fig 9G–9J, especially in the fovea region. Only the proposed method can generate topological and smooth results (Fig 9K). The same experiment was conducted on the data of all observers and the results are available in *S5 Text*. These results clearly show that the proposed method can generate smooth and topological retinotopic maps in V1.

## Retinotopic maps of the V1/V2/V3 complex

We took the initial visual area labels from the multimodal surface matching (MSM) results of the HCP group [57,58], removed phase-jumping and phase-reversal for the left and right hemispheres in multiple visual areas after polar angle transformations, and achieved topological and smooth results across them (Fig 10). Fig 10A–10E show the visual coordinates in the eccentricity-extended-polar coordinate space, in which the $x$ axis is the eccentricity, and the $y$ axis is the extended polar angle (Table 1), for the raw data, average smoothed data, median smoothed data, Laplacian smoothed data, and topological smoothed data, respectively. All smoothing methods reduced topology violations, compared to the raw data. However, only our proposed smoothing method generated topological mapping such that the contour curves (the green and blue curves in Fig 10K) were smooth and had no self-intersection. We performed the same comparison across all observers (Table A in S5 Text) and showed that our proposed topological smoothing was the best in topology preserving.

## Boundary delineation

In the eccentricity-extended-polar coordinate space, the boundaries between different visual areas in the V1/V2/V3 complex are defined by specific polar angles. Fig 11A shows the boundaries directly inferred from the raw data of the first observer. Fig 11B–11E show the boundaries generated from the smoothed data of the same observer, after average, median, Laplacian, and topological smoothing. The new boundaries largely coincided with manually labeled visual areas from the average data, represented by the colors of the cortical surface Fig 11). The green curves (from the left to the right) are eccentricity contours from 1˚ to 6˚ with a 1˚ interval, and

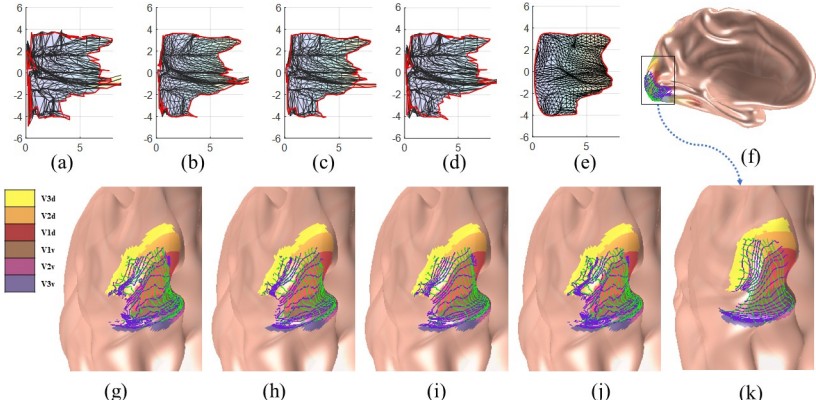

**Fig 10.** The retinotopic mapping of the V1/V2/V3 complex of the left hemisphere of the first observer: (A) the raw retinotopic map in the eccentricity-extended-polar coordinate space (with 232 flipped triangles), (B)-(E) the smoothing results of the Average (with 55 flipped triangles), Median (with 85 flipped triangles), Laplacian (with 151 flipped triangles), and the proposed smooth (no flipped triangles), respectively, (F) the entire left inflated mesh in medial view, (G) an enlarged graph of the raw retinotopic map on the inflated mesh, (H)-(K) smoothing results on the inflated surface of the four smoothing methods, in the same order of (B)-(E). In (A)-(E) the $x$ axis is the eccentricity $v^{(1)}$, and the $y$ axis is the extended polar angle $v^{(2)}$, in (H)-(K), the green and blue curves are levels sets, i.e., the contours of eccentricity $v^{(1)}$ and extended polar angle $v^{(2)}$, respectively.

the purple curves represent boundaries of visual areas (V3d/V2d/V1d/V1v/V2v/V3v, from up to down). As in the retinotopic maps, topological violations occurred in boundary delineation from the raw data and after smoothing with all but our proposed method. We present all observers' area boundaries after topological smoothing in *S5 Text*. They were all smooth and continuous.

## Discussion

In this study, we modeled the topological condition and generated topological and smooth retinotopic maps. The Beltrami coefficient, a metric of quasiconformal mapping, was used to define the topological condition. We developed a mathematical model to quantify topological smoothing as a constrained optimization problem and elaborated an efficient numerical method to solve it. The method was applied to V1, V2, and V3 simultaneously, and was robust to inaccurate boundary labeling. Experiments with both simulated and real retinotopy data demonstrated that the proposed method could generate topological and smooth retinotopic maps. As a result, we were able to improve boundary delineation. To our best knowledge, conventional methods have not fully considered topological constraints for multiple regions in retinotopic smoothing. Our novel algorithm made the retinotopic maps from BOLD fMRI topological and consistent with results from neurophysiology. Our work improved the quality of retinotopic mapping and built a solid foundation for future retinotopy related research.

### Quality assessment

In this study, we enforced the topological condition in retinotopic mapping. This enforcement could potentially introduce large visual coordinate changes to the raw data compared to other smoothing methods. It is important to monitor whether the smoothing results are acceptable or have relatively big deviations. Such quality assessment can provide the right level of confidence when drawing conclusions based on the smoothing results. In Fig 12, the visual coordinate changes of two typical observers are shown on the cortical and parametric surfaces respectively. Fig 12A shows the studied visual areas on the 3D cortical surface. Fig 12B and 12C shows the

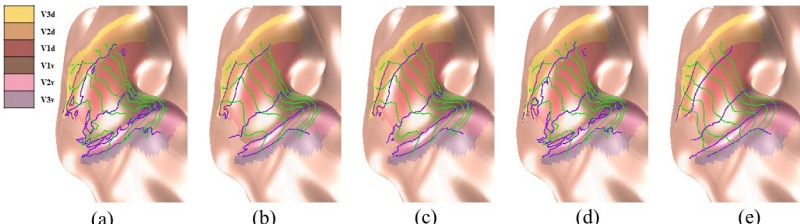

**Fig 11. Visual area boundary delineation of the V1/V2/V3 complex of the first observer.** The colors of the surface represent manually labeled visual areas from the average data. The purple curves indicate boundaries between visual areas of V3d/V2d/V1d/V1v/V2v/V3v (from up to down), respectively. The green curves are eccentricity contours. (A) boundaries inferred from the raw data, and (B)-(E) boundaries after average, median, Laplacian, and topological smoothing.

distortion in visual space. The colors indicate the amount of visual coordinate changes for the proposed smoothing process. The first observer's results (Fig 12A) suggested that the smoothing was generally good, with on average an about 1.12˚ change of visual coordinates. In contrast, the other observer's results (Fig 12C) suggested that some sub-regions have relatively big visual coordinate changes, with on average an about 1.75˚ change of visual coordinates. We also show the mean visual coordination distortion distribution of all 181 subjects in Fig 12D. Together with Table 5, the quality assessment results ensure our retinotopic mappings well respects the original fMRI signals.

## Out-most boundary condition

In this work, visual areas, V1/V2/V3, and the initial boundaries between them were determined by surface registration. Following the HCP protocol [36], MSM [58] was adopted to register each observer's cortical surface to the retinotopy atlas obtained from the HCP [34]. After the registration, the boundaries for each observer were determined along equal polar angle curves, and our extended polar angles were initialized with these boundaries. The out-most eccentricity boundaries were manually determined based on the eccentricity signal range (in our case, approximately from 0.2˚ to 7˚).

The initial boundaries might not be accurate since they were determined by surface registration. With extended polar angles, our algorithm was robust enough to fix the interior visual area boundaries. Our method requires that the visual coordinates of the out-most boundaries are specified when solving the LBS. Even so, if the visual coordinates of the out-most boundary are not properly given, we can still update the out-most boundaries and recover interior visual coordinates. We tested two approaches to set the out-most boundary values. One approach is to set a fixed boundary for all observers to simulate the case in which a good boundary configuration is given, and the other approach is to adjust the out-most boundary values by fitting a smooth spline of the interior visual values to simulate the case in which the boundary

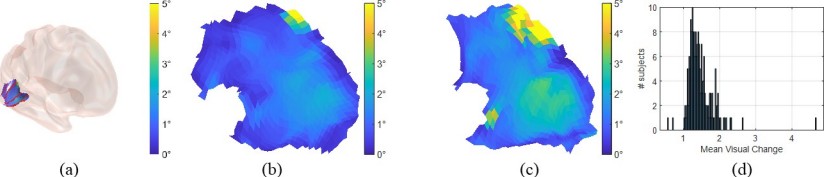

**Fig 12. Visual coordinate change after smoothing.** (A) The visual change is rendered on the inflated cortical surface (B) The visual change is rendered on the parametric surface for the first observer. Similarly, (C) is the visual change for the second observer. (D) is the histogram of average visual change for the HCP dataset (N = 181).

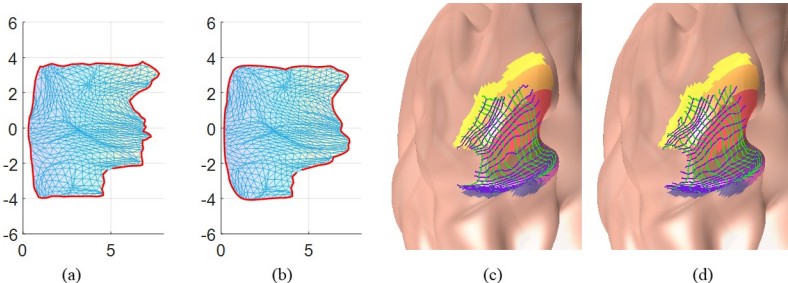

**Fig 13. Comparison of two boundary approaches.** (A) The result of fixed boundary visual coordinates. (B) The result of changeable boundary visual coordinates. (C) Result of (A) on the inflated cortex. (D) Result of (B) on the inflate cortex.

coordinates are given but not precise. Each approach has its own pros and cons. The first approach is numerically stable but ignores individual differences, while the second approach adjusts the boundary values based on retinotopic data, which has higher accuracy but may fail to set reasonable boundary values. We compared both approaches in the experiments. Fig 13A and 13C are the results from the first approach, and Fig 13B and 13D are the results from the second approach. We found a relatively large difference on the out-most boundaries (red boundaries in Fig 13A and 13B), while the interior regions were less affected. The mean difference between the results of the two approaches was 0.23˚±0.15˚, which is relatively small. In this paper, we took a combined approach: the boundary values were set within a small tolerance range (0.5˚) of the initial average boundary values. Nevertheless, we found that, although the out-most boundaries may not be precise, the interior results were not much impacted by the boundary values. Therefore, we can use the pre-defined boundary configuration for all the subjects.

It is worth noting that our method is applicable to the registration results from Freesurfer [37] or MSM [58] or Bayesian register [8]. Here we applied it to successfully process data from all 181 HCP observers. In our code repository, we have published all the necessary manuals and atlas for others to reproduce our work. We also provided a simple software tool to manually draw the out-most boundaries on visual cortical surfaces and generate corresponding visual coordinates.

## Robustness with inaccurate interior boundaries

When smoothing the retinotopic maps of the V1/V2/V3 complex, we applied linear transformations to the polar angles based on the visual region labels derived from the anatomy. Here, we explain why our proposed smoothing method was robust even with inaccurate boundary specifications.

We used the first observer's data as an example to illustrate that the proposed method was robust to inaccurate interior boundaries (Fig 14). In Fig 14A, we show the raw visual polar angle data on the parametric disk for the first observer, with a white curve at 0.3-degree eccentricity. Then we applied a linear transformation before smoothing, according to Table 1. For a vertex with an inaccurate boundary label, the polar angle would be transformed with a wrong offset. Since the transformation adds or subtracts multiples of $\pi$ as the offset according to its visual region label in Table 1, a wrong label makes the transformed value dramatically different from its neighboring value. Therefore, an incorrectly transformed vertex looks like a spike and destroys its topological condition (red points in Fig 14B).

In the next step, we applied the topological smoothing (smoothed visual polar angle is the slope-shaped surface in Fig 14B). Since it enforces topology conditions, our topological

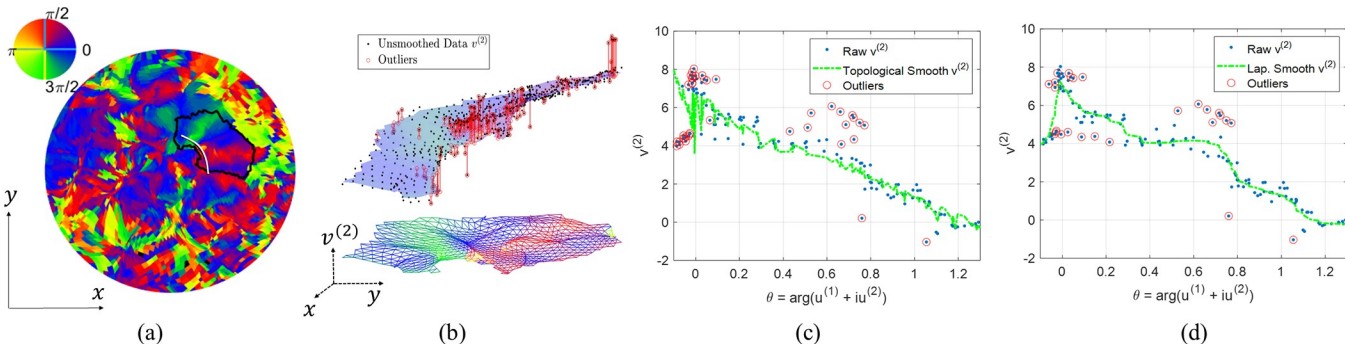

**Fig 14.** Smoothing is robust to the inaccurate interior boundary: (A) the raw visual polar angle $v^{(2)}$ is rendered on the parametric unit disk (i.e., radius is one), (B) $v^{(2)}$ as a function on the cropped region, (C) visual polar angle $v^{(2)}$ along the white arc by topological smoothing, (D) similar result as (C) but the curve is from Laplacian smoothing.

smoothing method replaces the spike with neighboring values. Namely, the proposed method can fix the wrong offset and is therefore robust to inaccurate interior boundaries.

We further verified that the robustness was not due to Laplacian smoothing. In Fig 14C, we plot the raw data and topologically smoothed results of the white curve. The white curve is formed by points with eccentricity $v^{(1)} = 0.3°$ with tolerance $0.05°$. We now consider the data near the white curve. If the eccentricity measurement is accurate, the 2D topological condition can be considered as a monotonicity requirement. In Fig 14C, the blue dots are the raw extended polar angle data, and the green curve is the topological smoothing result. Even though there were outliers (the red circled points, mainly due to the noisy eccentricity data), the main portion of the trend was monotonic. In Fig 14D, we plot the results with only Laplacian smoothing without topology constraints as the green curve, which is crooked in order to fit the data but not monotonic (topological). This means Laplacian smoothing can smooth but cannot fix topological violations. In contrast, our topological smoother ensures topological condition, which essentially makes the mapping "monotonic". It is the topological constraint rather than Laplacian smoothing that makes the smoothing more robust to inaccurate interior boundaries.

The new topology-projection module in the proposed method makes it robust to inaccurate interior boundaries. The smoothed extended polar angles from topological smoothing can be used to infer visual area labels.

We computed visual coordinate changes following an alteration of the V1/V2 boundary. If the visual coordinates did not change too much when the boundary was altered, the results were robust to interior boundaries. We started from the first subject's data (Fig 15A). We

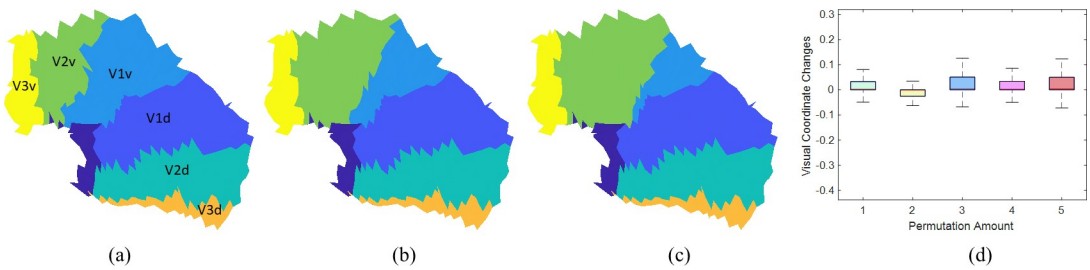

**Fig 15.** (A, B, C) Perturbations of the V1/V2 boundary, where (B) was obtained after three rounds of expansion and (C) was obtained after five rounds of expansion. (D) Visual coordinate changes with different degrees of perturbations.

expanded V2v by moving points in V1v into V2v if they were adjacent to V2v. The expansion was repeated to simulate different boundary misplacements (three rounds in Fig 15B and five rounds in Fig 15C). Then we applied our smoothing method and compared the results with no perturbation. The boxplot in Fig 15D shows that the visual coordinate differences were relatively small (the average is less than 0.5˚) with V2v boundary expansion. The results suggest that the proposed method is robust to interior boundary placement.

## The trade-off between smoothness and quality of fit

The smoothing weight $s$ in Eq (7) controls the trade-off between smoothness and quality of fit under topological condition. Although other studies have attempted to find the optimal $s$ for linear smoothers [59], the topology constraints made it challenging. We used the following way to set the proper $s$: find the largest weight while ensure that the average difference between the smoothed visual coordinates and the raw data is within a certain range (1˚ in this work). To empirically determine the $s$ value, we computed the percentage of the non-topological area and the Normalized-Root-Mean-Square-Error (NRMSE) of the fit to the fMRI time series for the raw data, when $s = 0$, and when $s$ was set properly for the first observer. For the raw data: $NRMSE = 0.07944 \times 10^{-3}$, 24.7% of the area is non-topological; when $s = 0$, $NRMSE = 0.08546 \times 10^{-3}$, 0% of the area is non-topological; and when $s = 0.0121$, $NRMSE = 0.8551 \times 10^{-3}$, 0% of the area is non-topological. The NRMSE values for $s = 0$ and $s = 0.0121$ were quite close. Therefore, we prefer the latter result in this work.

## Beltrami coefficient vs. Jacobian

In this work, the topological condition is ensured by enforcing $\|\mu\|_\infty < 1$. Another geometric concept, the determinant of the Jacobian matrix $J$ can also be adopted to enforce the topological condition. For example, it has been adopted by previous work to monitor the local visual field signs of retinotopic maps [14], which indicate the trend of visual polar angle changes (increasing or decreasing) on an iso-level eccentricity curve. Constraining the determinant of the Jacobian to be positive can also ensure the topological condition and visual field sign within a neighborhood. Mathematically, the Beltrami coefficient and the determinant of the Jacobian matrix are related via the following equation: $J = |f_z|^2 - |f_{\bar{z}}|^2 = |f_z|^2 (1 - |\mu|^2)$ [60].

We prefer the Beltrami coefficient because it is more suitable for retinotopic mapping. *First*, with the Beltrami coefficient, we can define the topological constraint as a penalty term to smoothing energy (Eq 7) because the Beltrami coefficient quantifies angle distortions and retinotopic mapping is considerably angle preserving [54,61]. In contrast, $J$ cannot be added as a term in smoothing energy because it quantifies area changes, and retinotopic mapping is not area-preserving. Numerically, one must separately treat the constraint and adjust visual coordinates to ensure that $J$ is postive. *Second*, as indicated by Theorem 1 in S2 Text, the Beltrami coefficient map uniquely determines a diffeomorphic mapping between two 2D surfaces, making it possible to recover/solve the mapping if we know the Beltrami coefficients. Therefore, we can divide the smoothing process into two subproblems and solve each separately and efficiently. To our knowledge, there are no efficient and stable methods to utilize Jacobian $J$ to reconstruct diffeomorphic mappings. In summary, although it may be possible to use $J$ to quantify the topological condition, the Beltrami coefficient is more suitable for modeling retinotopic maps.

## Smoothness and topological projection

The proposed topological smoother produces topological and smooth retinotopic maps. Although these two conditions are related, the topological condition does not ensure

smoothness, nor does smoothness ensure the topological condition. In fact, the topological condition can be ensured without smoothing. Empirically, no one expects retinotopic maps to be topological but extremely irregular and rough.

Smoothing does to some degree improve the topological condition. Although we do not know the exact relationship between visual coordinate smoothing and the topological condition (monitored by Beltrami coefficient), the Beltrami coefficient quantifies the local deformations based on its definition. Smoothing can reduce the outliers in Beltrami coefficients. Because 80% of the triangles in the HCP dataset are orientation preserving (See Table A in S5 Text), smoothing the visual coordinates may reduce the norm of the Beltrami coefficients in average and improve the topological condition. If the data have more flipped triangles (e.g., ~50% triangles' are flipped), smoothing may introduce more non-topological triangles and make the problem worse.

The topological condition cannot be ensured by reiterative smoothing. As shown in Fig 15D, repeated Laplacian smoothing did not generate fully topological results. The topological condition is mainly fulfilled by topological projection. However, after fixing the topological condition, a major concern is the results could be too irregular or deviate too much from the original pRF solutions. The proposed *Topological smoother* can produce results that are both topological and smooth. Roughly speaking, if the fMRI signal to noise ratio is very high (say SNR >20), the pRF solutions would be smooth and topological. For example, the pRF solution of the averaged fMRI signals from many subjects (https://osf.io/bw9ec/wiki/home/) has very few flipping triangles in V1. This is why the energy function in the topological smoother has a term that penalizes non-smoothness and therefore may improve the fit to the data and improve the topological condition. We believe the approach can balance the topological condition and smoothness.

## Extension to higher visual regions

Although we focused on the V1/V2/V3 complex in this study, our method can be in principle extended to higher visual areas. However, there are some additional challenges. As we move to higher visual areas, the signal-to-noise ratio of the retinotopic map is further reduced. It is extremely challenging for all smoothing methods to balance the topological condition and goodness of fit. As shown in Fig 16, we applied our proposed smoothing method on the V3B retinotopic maps of the first four observers in the HCP dataset. Instead of smoothing V3B with V1-V3 together, we performed separate smoothing on V3B and concatenated the results with those on V1-V3. Although topological smoothing was feasible in higher visual regions, extra care must be taken because of the low signal-to-noise ratio of BOLD fMRI activation.

## Benefits of the extended polar angles

Our previous work on topology correction [29] was only applicable to V1; it could not be used to improve boundary delineation. In this work, by introducing extended polar angles, we

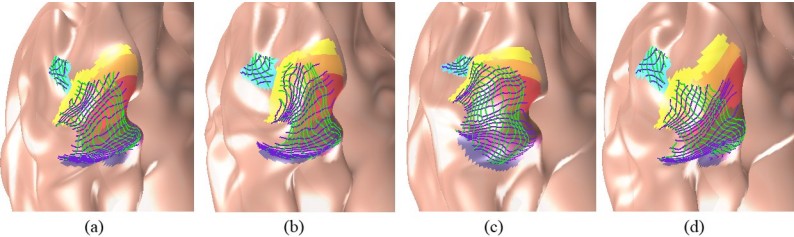

(a)          (b)          (c)          (d)

**Fig 16. Smoothing extended to the retinotopic map of V3B (the blue color region) of the first four observers.**

extended the framework to the V1/V2/V3 complex and achieved topology-preserving smoothing in multiple visual areas simultaneously. Although our previous work [29] might be potentially appliable to V2 and V3, accurate V1/V2 and V2/V3 boundaries were necessary. In the current work, given the out-most boundaries of the V1/V2/V3 complex, our work not only generated more precise smoothing within the interior of each region, but also automatically delineated the boundaries between them. The new development on boundary delineation may significantly improve the robustness and accuracy of retinotopic mapping. In addition, our prior work was mainly evaluated with visual coordinate the changes. How much impact it had on the goodness of fit of the fMRI time series was not evaluated. Here we added fits to the fMRI time series in performance evaluation, which may better justify our work and alleviate the concern that the proposed method may over-smooth the retinotopic maps. With extensive evaluation based on both synthetic data and real retinotopic map data from 181 HCP observers, the current study demonstrated that our proposed method may improve the accuracy and stability of retinotopic mapping, especially the interior boundary conditions.

### Elaboration on the reduced $R^2$ in Table 5

The topological smoother can completely remove topology violations in retinotopic maps. However, as shown in Table 4, it reduced the measure of explained variance, $R^2$. In general, the pRF model should have a better $R^2$, since $R^2$ values are computed at each point on the cortical surface. There are no constraints, nor penalty for non-topological results in $R^2$. However, neurophysiology studies strongly suggested that retinotopic mapping is topological for normal subjects. If the pRF results are not topological, the problem is very likely caused by the low spatial resolution and SNR of the fMRI data, an issue that cannot be addressed with current MRI technologies. In this work, we assume that the neurophysiology results are correct and attempted to fix topological violations in retinotopic maps by slightly updating the visual coordinates. We used pRF results as input data without 1) fine tuning pRF model parameters, such as the gain or the exponent of the power function [62], and 2) iterating with the pRF model to search for an optimal solution with balanced fitting quality and topological condition. Our ongoing work [62] tries to integrate the pRF model and topological conditions by defining a combined energy term. We expect such a balanced approach may improve topological condition without sacrificing data fitting quality.

### Relation to registration

The Beltrami coefficient (BC) encodes 2D-to-2D mapping up to a normalization [60]. Manipulating the Beltrami coefficients of the registration function between a subject and a template can also achieve diffeomorphic registration and generate topological results [63]. The benefits of this approach was discussed in [8]. However, one main issue is that the template itself might be biased. One potentially promising solution is to combine registration and smoothing: while registration provides good boundaries, topological smoothing provides good interior smoothing [62].

### Conclusion and future work

We adopted the Beltrami coefficients to quantify the topological condition in retinotopic mapping, formed a concise but fundamental model, and provided an efficient numerical solver to generate smooth retinotopic maps. To our knowledge, the proposed topological smoother is the first method that guarantees the topological condition in retinotopic mapping. Our work improved the quality of the retinotopic maps and built a solid foundation for future research based on retinotopy.

## Supporting information

**S1 Text. Notations and definitions.**
(DOCX)

**S2 Text. Beltrami coefficient.**
(DOCX)

**S3 Text. Discrete operators on a 2D triangular mesh.**
(DOCX)

**S4 Text. Angle Distortion.**
(DOCX)

**S5 Text. Supplementary data and code.**
(DOCX)

## Acknowledgments

The first author would thank Dr. Chengfeng Wen of Stony Brook University for providing fundamental geometry processing packages at the beginning of this project.

## Author Contributions

**Conceptualization:** Yanshuai Tu, Duyan Ta.

**Funding acquisition:** Yalin Wang.

**Investigation:** Yanshuai Tu, Duyan Ta.

**Methodology:** Yanshuai Tu, Zhong-Lin Lu, Yalin Wang.

**Project administration:** Zhong-Lin Lu, Yalin Wang.

**Resources:** Zhong-Lin Lu, Yalin Wang.

**Software:** Yanshuai Tu, Yalin Wang.

**Supervision:** Zhong-Lin Lu, Yalin Wang.

**Visualization:** Yanshuai Tu.

**Writing – original draft:** Yanshuai Tu.

**Writing – review & editing:** Zhong-Lin Lu, Yalin Wang.

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
