## [Decision Letter · Decision Letter 0]

27 Jan 2021

Dear Dr Wang,

Thank you very much for submitting your manuscript "Topology-Preserving Smoothing of Retinotopic Maps" for consideration at PLOS Computational Biology.

As with all papers reviewed by the journal, your manuscript was reviewed by members of the editorial board and by several independent reviewers. In light of the reviews (below this email), we would like to invite the resubmission of a significantly-revised version that takes into account the reviewers' comments.

Please particularly pay mind to points on clarity and validation. As Reviewer 3 points out there is a previous short conference paper, which should be cited and extensions relative to that work should be clarified.

We cannot make any decision about publication until we have seen the revised manuscript and your response to the reviewers' comments. Your revised manuscript is also likely to be sent to reviewers for further evaluation.

Sincerely,

Emma Claire Robinson

Associate Editor

PLOS Computational Biology

Wolfgang Einhäuser

Deputy Editor

PLOS Computational Biology

Reviewer's Responses to Questions

**Comments to the Authors:**

Reviewer #1: The paper presents a method for retinotopic maps that preserve topology. Beltrami coefficent is used to define the topology.

* Why use Beltrami coefficent, surface Jacobian can be computed directly and that could be a more direct measure of distortion. In fact surface Jacobian has been used by the authors in the past.

* More description of Beltrami coefficent should be added to the paper.

* How does the the Beltrami coefficent relate to the topology and distortion should be described in more detail possibly with some illustrations.

* How does the smoothing in fMRI data affect the Beltrami coefficent?

* I think the notation of first fundamental form in eq 3 might not be necessary. If the authors want to keep it, more description of them should be added.

Overall, this is a very interesting work. Addressing the above issues would improve the paper.

Reviewer #2: The manuscript by Tu, Ta, Lu & Wang introduces and develops a new method for analysing retinotopic maps. Essentially, the method enforces a topological mapping from retina/visual field space to cortical space, by using the Beltrami coefficient. Turning a measurement of many independent and noisy estimates into a consistent mapping bears many advantages and this has been previously recognized by Doughtery et al 2003 and Schira et al. 2007, using a technically different approach of morphing a map template on retinotopic data. The procedure suggested here has many advantages and some disadvantages over previous methods. The approach suggested seems very elegant and promising, the implementation points to great skills, and the method could indeed significantly improve our analysis of retinotopic maps. However, I am struggling with the current manuscript both in clarity but also structure. I suggest the authors make the purpose clear and structure the manuscript accordingly. Importantly, measures and analysis such as CMF or quality assessment need to be significantly improved or removed from the manuscript. At present they seem like an afterthought and that does neither these assessments nor the method justice.

1. The authors should do a better job explaining their core assumption that is that the mapping between retina/visual field and cortex is topological. I feel Fig 1 is ill suited and instead confusing in this context. Upon reading Fig 1 (for example F1b and F1c) I was trying to understand what topological concept the six nodes and eight edges in F1c violate, for example they could form a handle or a hole in the surface. However, that little set of triangles is a perfectly fine surface if rotated. F1d is really not adding anything either. Essentially the concept is that a sequence of locations on the retina, such as a,b,c are represented in this order on cortex and not for example as a,c,b. This is obviously hard to illustrate in a figure without retinal space. The same applies to F2, where I am not certain what it explains.

2. The project starts with the proposal that no assumptions are needed other than enforcing the “topological condition”, but unfortunately this is not exactly correct. In order to map across several visual areas, the accuracy of boundary labels is a major concern. An ill placed boundary can significantly impact the smoothing process, by introducing apparent breaches in topology that do not exist in the data. The section “Robustness with inaccurate interior boundaries” L664f tries to address this, but the section is entirely insufficient and has a number of flawed assumptions. I was also struggling with comprehension of this section.

3. Specifically, the authors imply that their method is robust against such errors, but it is entirely unclear what they mean by this. Will mislabelling a) simply be “smoothed over without larger consequences” or if b) the method somehow corrects these mistakes and instead finds the most correct boundary. I suspect it is a). If it is indeed b) the mechanism by which this is done needs to be clearly specified. In fact it is unclear how area labels are created in the very first place.

4. Assuming it is a) error will inevitably distort/smooth surrounding mapping, confounding attempts to quantify local CMF. It would be important to quantify these distortions, for example by systematically moving internal boundaries such as the V1/V2 border by medium to small amounts (15mm?) and investigate and quantify the effects.

5. Would it be possible to conceive a method that optimises the boundary placements? Maybe even use the Beltrami coefficient to find the optimal boundaries?

6. It remains unclear throughout the entire manuscript what Software (Language) the authors used for implementation. I had to go to the OSF site of the project to find matlab code. Searching the document matlab is cited as reference no 51, but [51] is actually not cited in the text.

7. The authors apply their method to datasets of 171 participants, but for what purpose? Only the results in Fig. 13 seem to rely on 171 datasets. It is a curious result but it breaks the consistency of the manuscript that otherwise seems to be a pure method paper.

8. The authors describe their technique as topological smoother. To me the two things topological correctness and smoothness are not necessary the same. The name seems to suggest the method smoothes until the topological condition is met. Does that mean datasets with large inconsistencies end up being more smoothed?

9. I was majorly struggling (failing) to follow large sections of the manuscript, in particular pages 11 to 16. I felt the authors are using a large number of symbols often inconstant and the purpose is not clear. I understand that it is hard to structure a logical problem that arose piece by piece when implementing a new method, but I feel the authors need to significantly improve this. I am unclear what the first subproblem exactly is, and what EXACTLY is smoothed. I understand what Laplacian smoothing is, but I am unclear what the mesh is that is smoothed.

10. I am unclear how it is the second subproblem is solved. Do I understand correct that nodes with negative coefficients are found, then their coefficient is changed by “shrinking” whatever that means. Shrinking sounds like an iterative process. L337 refers to Figure 2d which does not exist. I believe it should point to F2b red. Also the reference to fi seems wrong, fi denotes the node most to the right and about halfway up the positive axis. The red node does not seem to have a designation. How is that linked with the first subproblem?

11. L560F “angle distortion”. Angle distortion is typically called anisotropy, and yes, this has been “formally formalized”. It is an extremely common topic in the field of vision science with many authors discussing and analysing it, even Holmes and Lister (1916) were discussing it. The pattern anisotropy at eccentricities 0.5 and 1 degree displayed in Fig. 13 looks suspiciously like the one found by Schira et al. (2007, 2009 and 2010). The asymmetry for eccentricity ranges 1.5 and 2 degrees (yellow and purple curve) however has not been reported before. However, if the authors would like to present these results, they need to deliver a much more solid report on how it was obtained, they need to provide an analysis of the variance (i.e. error bars…) across subjects and make sure the results are not the consequence of a systematic error. Just glancing at these I would be very suspicious the anisotropy found at -pi/2 (ventral?) but not at pi/2 (dorsal?) could be a systematic error, as nothing like this has ever been reported. Extraordinary claims require extraordinary evidence.

12. L39 “The violation of topology condition is especially severe near the fovea, “ this remains a little fuzzy, not specified what ‘near the fovea’ means, like 1 degree eccentricity?

13. L62 “recent advances in Blood Oxygenation Level Dependent…” This statement feels a little bit out of time. These “recent” advances have allowed retinotopic mapping and estimation of magnification more than 25 years ago by several authors (Sereno et al, De Yoe et al, Engel et al.) “recent advances” is a phrase commonly used to highlight sub mm fMRI, and even in this field the phrase feels dated. This somewhat extends to the entire paragraph. In the last two decades many more visual areas than just V1, V2 and V3 have been discovered.

14. L120 suggests fi is “a mapping”. I find this confusing and suspect it’s wrong. I also do not understand how fi can be “moved out of any of the edges“ (L121). The way it looks to me, in F1c the edge fjfk (or the triangle fjfkfi) violates the surface. The authors really need to clarify their terminology here.

15. I suspect the authors may understand F1 as an illustration of a surface to surface mapping, where F1b) is the retina and F1c) is the cortical surface. Then for the node fi the topology of the mapping is violated. However, then F1a) is confusing and the entire Figure 1 does not consider the retina a part of the mapping. Then the triangle PiPjPk is in the retina/visual space and the tringle fifjfk in cortex?

16. L194 ”assuming that the voxel’s respective is (’;,)” what is a respective? I am not certain it is required for this manuscript to describe the equations of the pRF fitting procedure. A simple 4-5 line section with quoting the relevant manuscript describing the specific implementation used should suffice.

17. L291 “we now explain how to recovery…” should be “recover” I think.

18. L 595 “Quality assessment”. I am not certain what that paragraph is supposed to illustrate, other than apparently that for S1 the method changed the map only a little, while for S2 it changed a lot. What does that mean? What about the other 169 subjects? What it too much what is too little? Of course, quality control is an important step in any data processing/analysis and hence a worthwhile discussion topic. The authors should try and define some sort of criteria/logic. For example the authors could provide a quantitative measurement of the amount of smoothing and report how these values are distributed. This would allow to classify good vs bad datasets and provide a framework to judge this. One could argue this would go beyond the scope of the publication, but if included it must be of substance.

Reviewer #3: This is a method manuscript. It tackles a very specific problem in population receptive field (pRF) fitting algorithms. Basically, most of the pRF algorithms work at the voxel level, and per every voxel, the algorithm will provide the pRF center and size. By definition, those pRF centers need to be unique and retinotopically organized. Usually, these voxels are mapped to the surface, in a 2D surface mesh, so that every vertex corresponds to a time series. As these fMRI time series are very noisy, it can happen (in 20% of the cases according to the authors) that the pRF center values estimated from those time series break the topological condition. The authors propose a method to solve this issue, generating topology-preserving retinotopic maps.

pRF algorithms are very powerful but they are relatively modern (they were introduced in 2008). There are improvement opportunities, and it is important for the field that the authors identify a weakness of the method and propose a solution to it. The paper is clearly written, it is well organized and the figures are generally clear, which is a great pleasure as a reviewer.

Major comments

- The authors published this method already in ISBI 2020 (https://www.ncbi.nlm.nih.gov/pmc/articles/PMC7406191/#!po=43.3333). I think that: (1) If the method has been significantly improved, the ISBI paper should be cited and here explain what those improvements consist of. (2) If there are no improvements on the methods but there are in the applications (CMF and angle distortion quantifications?, discussion, examples...), the ISBI paper should be cited for the methods part and add the new contribution here. I can see how much work went into this manuscript, but it is the job of the editors to decide if the contribution is big enough to be published in their journal, but as a reader I would expect the other paper to be acknowledged/explained in this one.

- Validation of the method:

--- Synthetic data: If I understand it correctly, you basically take a topology and map an invented set of pRF center parameters, with more or less noise. I don’t understand why is the topology cuadricular when the vertices in the cortex are organized in triangles. I guess they are “equipotential” (angle, eccen) lines and not the triangles of the surface?

------ How can you prove that those modifications are realistical? I think that it would make more sense simulating time series with known ground-truth values (using Lerma-Usabiaga 2020 Plos Comp. Biol., for example) and adding noise until 20% of the triangles are flipped. And then applying the correction methods. You would know the ground truth of the parameters to estimate and therefore quantify the correction of your method.

------ How can you know that typically 20% in an experimental dataset are flipped? Table S2?

--- Experimental data: I don’t think the R2 calculations are the adequate ones here… Are you basically saying that for the time series, the solver went to a local minima but that there was a better solution, and that your algorithm could find the better fit to the time series? Knowing how most of the algorithms work ( seed > grid search > fine search), this can happen, but maybe there are better ways to show this than the one you provide. And if they are not, the concept of but you think it is going on should be explained, I think.

------ How can we know that the fixed data is closer to the truth? We are modifying the estimated parameters according to one criterion: we don’t want to have more flipped triangles. Do we need to assume that the averaged improved R2s are saying that we are closer to the truth or we are only saying that now we don’t have flipped triangles only?

- Boundary definition: do we need to set the boundaries manually to initialize the method or this is only necessary for the tests in the paper? I think it would be very useful for the reader/final user that you would add a small “user guide”. I understand that the user continues with exactly the same pipeline he/she was using, and then this method is something that is applied on top of it. After reading the manuscript, the manual and automated, the old and the new steps are not perfectly clear.

Other comments

- pRF Methods are not explained sufficiently: it seems from the text, as Dumoulin and Wandell 2008 is cited (Page 8, Line 168: Page 9, Line 192), that the authors use mrVista software to fit the time series, but in a second lecture, it seems that they use the same algorithm used in the HCP 7T publication, which I guess is Kendrick Kay’s AnalyzePRF?. I think that the software used should be explained specifically and maybe linking to publications/software repositories and versions, for clarity and reproducibility.

- There are no papers cited later than 2018, but there have been relevant publications in the field not acknowledged by the authors (for example Lage-Castellanos 2020, or the previously cited Lerma-Usabiaga 2020). In the same line, I missed discussing other approaches to pRF fitting, such as the paradigm free approaches (Merkel 2018, 2020).

- Page 2, Lines 44-45: that sentence is not correctly constructed.

- Ref 21 has an error it seems.

**Have all data underlying the figures and results presented in the manuscript been provided?**

Reviewer #1: Yes

Reviewer #2: Yes

Reviewer #3: Yes

PLOS authors have the option to publish the peer review history of their article (what does this mean?). If published, this will include your full peer review and any attached files.

Reviewer #1: **Yes: **Anand Joshi

Reviewer #2: **Yes: **Mark M. Schira

Reviewer #3: No
---

## [Decision Letter · Decision Letter 1]

27 Jun 2021

Dear Dr Wang,

We are pleased to inform you that your manuscript 'Topology-Preserving Smoothing of Retinotopic Maps' has been provisionally accepted for publication in PLOS Computational Biology.

Before your manuscript can be formally accepted you will need to complete some formatting changes, which you will receive in a follow up email. A member of our team will be in touch with a set of requests. You may use this as an opportunity to make the minor change to figure 1, suggested by R2, if you wish.

Best regards,

Emma Claire Robinson

Associate Editor

PLOS Computational Biology

Wolfgang Einhäuser

Deputy Editor

PLOS Computational Biology

Reviewer's Responses to Questions

**Comments to the Authors:**

Reviewer #2: Up reading the review of “Topology-Preserving Smoothing of Retinotopic Maps” by Tu, Ta, Lu and Wang, I find that the authors made a genuine and successful effort to improve and clarify the manuscript. The new section on robustness to inaccurate internal and external boundaries is useful and interesting. I find the manuscript is now ready to publication. The authors have addressed most of my concerns, except perhaps for Figure 1. For Figure 1 I would suggest the authors replace the brain background in F1a by a retina or eye background. This would make it intuitively clear that it is the estimated projection between retina and cortex that violates the topology condition. One may add the cortex picture as background to F1b), but that may be to much.

Reviewer #4: The authors present a very intriguing case for preserving the topology of population receptive field maps after standard parameter estimation methods. This is a fairly novel concept, in that location parameters of population units are being optimized for the topological condition based on their neighboring units in 2-dimensional vertex space via Beltrami-optimization.

The authors did respond to the reviewers concerns in great detail. After examining the initial submission and the comments to the initial reviews I remain slightly concerned about potential impacts of the ultimate enforcement of the topological condition on the location maps of the pRFs and their relation to the ground truth: The authors are right in their initial assumption of the neurophysiological reality of the topological mapping within retinotopic areas. However, the adoption of this method to the HCP-dataset shows, that its ultimate enforcement on the population level comes with the cost of a reduced raw-data-fit. The Beltrami-optimization does only change the location parameters of the initial pRF-solution (a solution estimating location and size(sd)) in order to retain topology. The R^2 between the raw-timeseries and the Beltrami-optimized time-series has to be lower than the initial pRF-estimated timeseries (as shown by the authors). I see now the problem of inferring whether this reduction in explained variance is a consequence of the change in location parameters OR keeping the size parameter constant. Sort of an answer can be found in the ground truth analysis, in which the authors did not take RF size into account. Since R^2 does not change for those synthetic datasets, one can assume that ignoring RF size in the topological optimization process may distort the estimated maps relative to the ‚real‘ topography.

One important next step would be to test ground truth incorporating all parameters used for the adopted pRF model.

The current topology-retaining smoothing method is certainly of interest for the community but more extensive simulation studies are required to test its validity.

**Have the authors made all data and (if applicable) computational code underlying the findings in their manuscript fully available?**

Reviewer #2: Yes

Reviewer #4: Yes

PLOS authors have the option to publish the peer review history of their article (what does this mean?). If published, this will include your full peer review and any attached files.

Reviewer #2: **Yes: **Mark M. Schira

Reviewer #4: **Yes: **Christian Merkel

---

## [Editor Report · Acceptance letter]

21 Jul 2021

PCOMPBIOL-D-20-02101R1 

Topology-Preserving Smoothing of Retinotopic Maps

Dear Dr Wang,

I am pleased to inform you that your manuscript has been formally accepted for publication in PLOS Computational Biology. Your manuscript is now with our production department and you will be notified of the publication date in due course.

With kind regards,

Andrea Szabo
